# Dynamic characteristics and fatigue life analysis of the gyration platform of roadheader

Shan Gao[1], Zhen Tian[1,2,3]*, Wei Liu[1], Lijuan Zhao[3], Yang Ge[1], Quan Sun[1]

1 College of Mechanical and Electrical Engineering, Zhoukou Normal University, Zhoukou, China, 2 College of Mechanical and Power engineering, Henan Polytechnic University, Jiaozuo, China, 3 College of Mechanical Engineering, Liaoning Technical University, Fuxin, China

* lntutian2008@126.com

**Data Availability Statement:** All files are available from the ScienceDB database (https://www.scidb.cn/anonymous/Sm5NYjly DOI:10.57760/sciencedb.14526).

## Abstract

In order to investigate the dynamic characteristics and fatigue life of the roadheader's gyration platform during the cutting process, the calculation of the cutting head load was achieved based on the force and its transformation of cutting pick, and the transient loads in the two lateral swing directions of the roadheader were calculated. Subsequently, the simulated load was applied to the multi-body model of the cutting section, and dynamic simulations were conducted. Critical information, including stress and strain distribution, as well as the maximum stress on the gyration platform, was extracted from these simulations. The study reveals that during the left-to-right cutting of the rock, the gyration platform experiences significant stress, with the high dynamic stress focus primarily concentrated at the bolt holes connected to the rotary bearings. Deformation was predominantly observed on both sides of the lifting cylinder and the rear portion of the gyration platform. Furthermore, both the maximum stress and deformation of the gyration platform exhibit a noticeable increasing trend with the escalation of swing speed. The correlation established indicates that elevated swing speeds lead to heightened dynamic stress on the gyration platform, consequently causing a noteworthy decrease in its fatigue life. To ensure the reliability of the gyration platform, it is crucial to judiciously select the swing speed based on the hardness of the rock, especially when cutting in different swing directions. The research results have important reference value for improving the structure and performance of the gyration platform.

## 1. Introduction

Excavation and mining represent indispensable procedures in coal production, where the guiding principle of "prioritizing mining while concurrently emphasizing excavation, with a preference for excavation" is commonly embraced. The expeditious and secure excavation of coal mine roadway stands as a fundamental prerequisite for guaranteeing elevated coal production [1,2]. In the demanding context of coal mine roadway excavation, the cutting heads

**Funding:** This work was supported in part by The Science and Technology Tackling Key Project of Henan Province under Grant 232102321097, and in part by Colleges and Universities Scientific Research Projects of Henan Province under Grant 25A440008 and in part by The Science and Technology Research Project of Zhoukou city under Grant 2022GG01008. The funders had no role in study design, data collection and analysis, decision to publish, or preparation of the manuscript.

confront a milieu of complexity and harshness. This is exacerbated by the unstable physical and mechanical attributes of the cutting target (a heterogeneous coal-rock mass). Consequently, the cutting heads are subjected to intricate and oscillating impact loads [3,4]. The pronounced impact forces frequently result in the abrasion and depletion of cutting picks mounted on the cutting heads, subjecting essential components of the roadheader, such as the gyration platform, to elevated dynamic stress. This dynamic stress can induce fatigue damage to critical elements, adversely affecting the regular course of excavation operations.

Many experts and scholars have conducted research on the dynamic characteristics of roadheaders. For example, Cheluszka conducted a comprehensive analysis of the fatigue life of conical picks during rock cutting, the study elucidated the factors influencing pick fatigue life and introduced an innovative design of cutting pick [5]. Zeng investigated the impact of cutting angle, cutting depth, and speed on the peak cutting force and cutting energy of pick, and conducted a meticulous analysis of factors influencing the fatigue life of conical pick [6]. Zhang contributed to the field by establishing a dynamic characteristic model for excavation machines through numerical simulation analysis. The work involved the study of vibration characteristics under diverse excavation conditions, with subsequent experimental validation [7]. Zhang employed finite element analysis to scrutinize the vibration of the gyration platform under various working conditions, and verified the finite element simulation results by single point excitation modal experiment [8]. Huang studied the dynamic characteristics of the coupling system under time-varying loads by establishing a cutting head rotor coupling system, and analyzed the effects of rock hardness, cutting speed, and rotational speed on the vibration of the cutting head [9]. Dogruoz conducted full-scale cutting experiments to analyze the cutting forces of cutting picks with different degrees of wear during the cutting process, and established a relationship model correlating wear and cutting force [10]. Liu established a multi-degree-of-freedom dynamic model for horizontal cutting of roadheader and analyzed the vibration characteristics of cutting heads and gyration platform through the multibody dynamics simulation [11]. Gao performed static analysis using finite element analysis software on the gyration platform of an EBH350 roadheader and proposed an optimized design scheme for the gyration platform [12]. Qu conducted vibration tests on the shuttle car of a roadheader, obtaining vibration acceleration and displacement data for multiple measurement points on the gyration platform, and analyzed the frequency and damping ratio of the gyration platform under different modal parameters [13]. Cheluszka analyzed the main factors that affect the cutting performance of a roadheader and established a mapping relationship between these factors and the mining energy consumption [14]. These studies have confirmed the accuracy and reliability of finite element and dynamic simulation results, yielding valuable outcomes. For the roadheader, the gyration platform as a vital component, linking the main body frame and supporting the cutting arm. During the cutting process, the gyration platform needs to achieve the lifting and rotation of the cutting arm and bear strong alternating loads, which is a weak link. However, few of the existing studies have involved the dynamics of the gyration platform under various lateral swing conditions and the influence of the lateral swing speed on the fatigue performance.

This paper conducted a comprehensive analysis of cutting pick forces during the excavation process of the EBZ75 roadheader. It conducted simulations and calculations to determine the loads on the cutting head when the roadheader operated in two distinct transverse swing directions. These loads were subsequently applied to a rigid-flexible coupling multibody model of the cutting section for dynamic simulation. The simulation yielded insights into the dynamic stress and strain distribution of the gyration platform under varied transverse swing conditions. Furthermore, the study delved into the fatigue performance of the gyration platform, establishing a relationship between the fatigue life and swing speed. The research results

provide valuable insights for strengthening the structural design of the gyration platform and the selection of appropriate swing speeds.

## 2. Materials and methods

### 2.1 Calculation of cutting head load

**2.1.1 Load analysis of cutting head.** The cutting head is composed of a head body, spiral blades, cutting picks, and pick holders, as illustrated in Fig 1. The main objective of the cutting pick is to excavate rock formations. When the roadheader engages in cutting rock with a cutting head rotational speed of $\omega$ and a transverse swing speed of $v_H$, the cutting picks experience forces as illustrated in Fig 2 [15,16]. These forces include cutting resistance $Z$, traction resistance $Y$ and lateral force $X$.

The cutting resistance $Z$ could be obtained:

$$Z = p_k[k_T \cdot k_g \cdot k_y(0.25 + 1.8t \cdot \sin\theta \cdot h) + 0.1s_z] \tag{1}$$

The traction resistance $Y$ could be obtained:

$$Y = k_{n1} \cdot Z \tag{2}$$

The lateral force $X$ could be obtained:

$$X = k_{n2} \cdot Z \tag{3}$$

In the formula above: $Z$ is the cutting resistance of a cutting pick, in N; $Y$ is the traction resistance of a cutting pick, in N; $X$ is the lateral force of a cutting pick, in N; $p_k$ is rock contact strength, in MPa; $k_T$ is the cutting pick type coefficient; $k_g$ is coefficient of influence on pick geometry, for conical pick: $k_g = k_\psi \cdot k_{\psi'} \cdot k_d$, $k_\psi$ is the shape coefficient of the alloy head, $k_{\psi'}$ is the

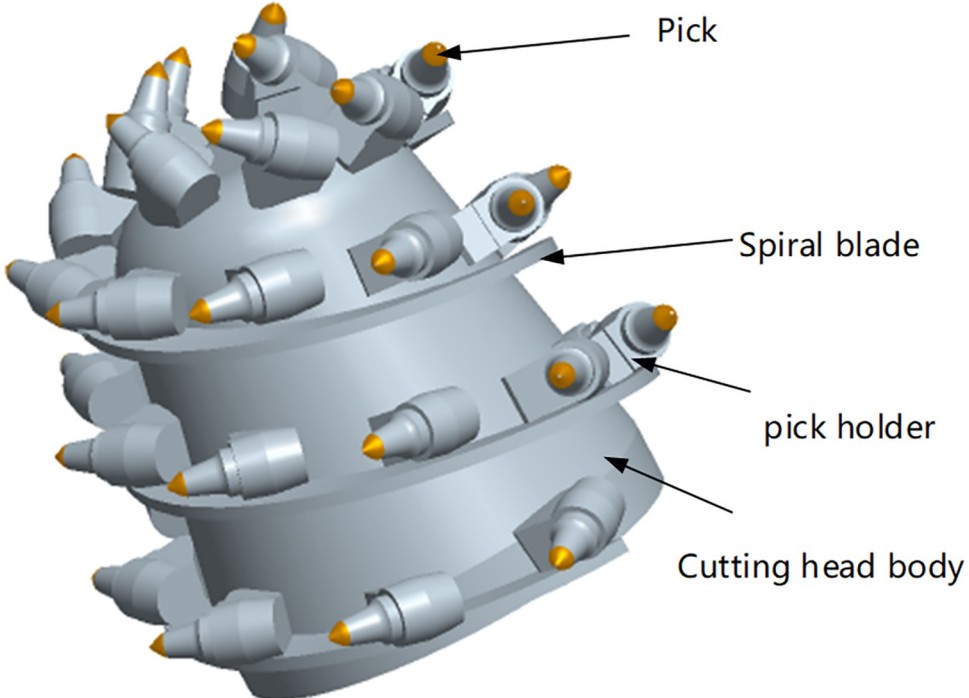

**Fig 1. 3D model of cutting head.**

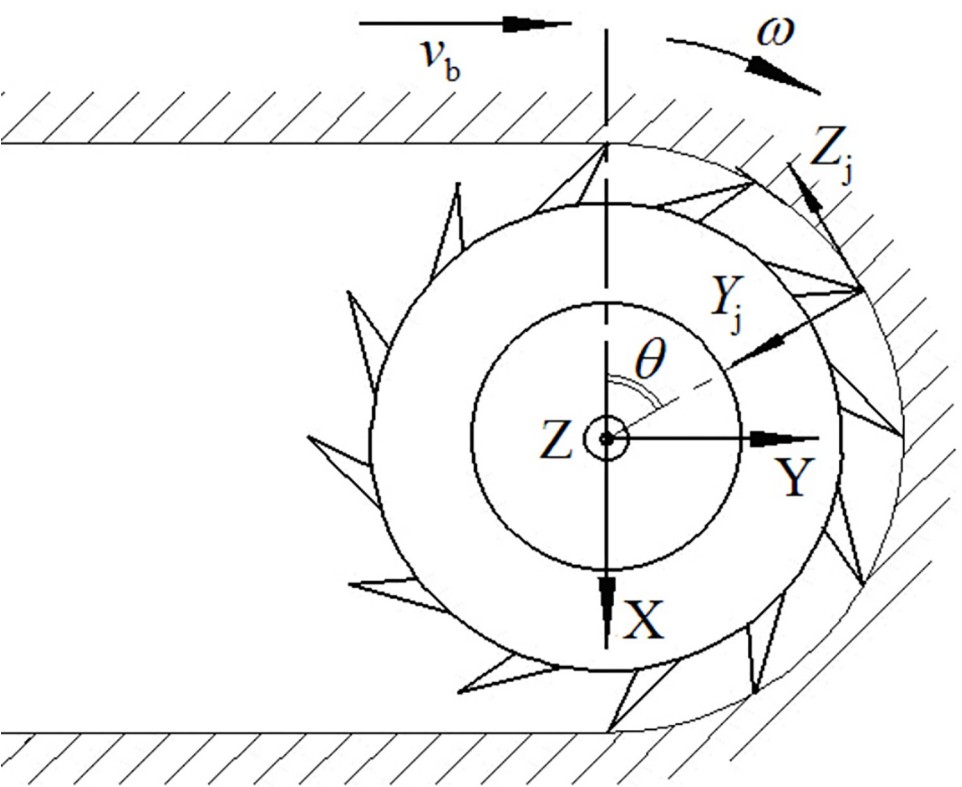

**Fig 2. Force analysis of cutting pick.**

shape coefficient of the cutting pick head, $k_d$ is the influence coefficient of alloy head diameter, for flat pick: $k_g = k_q \cdot k_b$, $k_q$ is the influence coefficient of cutting pick front edge shape, $k_b$ is the influence coefficient of the blade width of the flat pick, $k_b = 0.92+0.01b$, $b$ is the width of the pick shank, in mm; $k'_y$ is the influence coefficient of the cutting angle of the cutting pick; $t$ is the mean value of intercept distance, in mm; $h$ is the average value of chip thickness, in mm; $s_z$ is the projected area in the traction direction after blunting the rear edge surface, in mm$^2$; $k_{n1}$ is the coefficient of influence of cutting resistance on traction resistance; $k_{n2}$ is the coefficient of influence of cutting resistance on lateral force.

Upon computing the forces acting on each individual cutting pick, the forces exerted by all the cutting picks engaged in the cutting process can be converted to the centroid of the cutting head through coordinate transformation, as depicted in Fig 3. The three-dimensional resultant forces and resultant moments of the cutting picks on each cutting line of the cutting head can be calculated using Formulas (4) to (6) [17].

The resultant force in the X direction on each cutting line could be obtained:

$$R_{xi} = \sum_{j=1}^{n}(Y_{ij}\cos\theta - Z_{ij}\sin\theta) \tag{4}$$

The traction resistance in the Y direction on each cutting line could be obtained:

$$R_{yi} = \sum_{j=1}^{n}(-Y_{ij}\sin\theta - Z_{ij}\cos\theta) \tag{5}$$

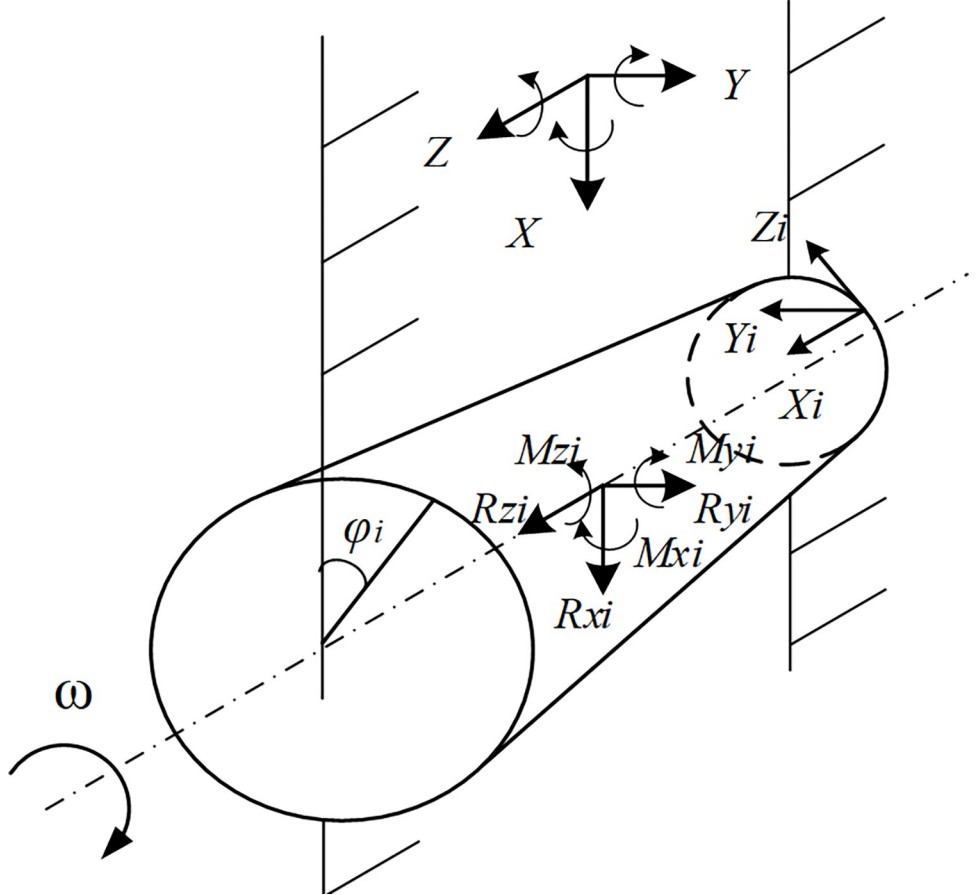

**Fig 3. Force transformation of cutting head.**

The lateral force in the Z direction on each cutting line could be obtained:

$$R_{zi} = \sum_{j=1}^{n} X_{ij} \tag{6}$$

In the formula above, $R_{xi}$ is the resultant force of the cutting picks on each cutting line, in N; $R_{yi}$ is the traction resistance of the cutting picks on each cutting line, in N; $R_{zi}$ is the lateral force of the cutting picks on each cutting line, in N; $i$ is the $i$-th section; $j$ is the $j$-th pick on each section line; $n$ is the total number of picks on each section line; $N$ is the number of cut lines; $X_{ij}$ is the lateral force of the $i$-th pick on the $j$-th section line, in N; $Y_{ij}$ is the traction resistance of the $i$-th pick on the $j$-th section line, in N; $Z_{ij}$ is the resultant force of the $i$-th pick on the $j$-th section line, in N; $\theta$ is the position angle of the pick, in °. Convert the force on all cutting picks on each cutting line to the centroid of the cutting head, and the resulting three-dimensional force and torque can be computed using Formulas (7) to (12).

The resultant force of the cutting head in the X direction could be obtained:

$$R_x = \sum_{i=1}^{N} R_{xi} \tag{7}$$

The traction resistance of the cutting head in the Y direction could be obtained:

$$R_y = \sum_{i=1}^{N} R_{yi} \tag{8}$$

The lateral force of the cutting head in the Z direction could be obtained:

$$R_z = \sum_{i=1}^{N} R_{zi} \tag{9}$$

The moment of the cutting head in the X direction could be obtained:

$$M_x = \sum_{i=1}^{N} R_{yi} \cdot d_i + \sum_{i=1}^{N} 500 R_{zi} \sin\theta \cdot D + \frac{3}{4} R_s \cos\alpha \cdot D \cdot 500 \tag{10}$$

The moment of the cutting head in the Y direction could be obtained:

$$M_y = -\sum_{i=1}^{N} R_{xi} \cdot d_i + \sum_{i=1}^{N} 500 R_{zi} \cos\theta \cdot D + \frac{3}{4} R_s \cos\alpha \cdot D \cdot 500 \tag{11}$$

The moment of the cutting head in the Z direction could be obtained:

$$M_z = \sum_{i=1}^{N} \sum_{j}^{n} Z_{ij} D \cdot 500 \tag{12}$$

In the formula above: $R_x$ is the resultant force of the cutting head, in N; $R_y$ is the traction resistance of the cutting head, in N; $R_z$ is the lateral force of the cutting head, in N; $M_x$ is the moment of the cutting head in the X direction, in N·mm; $M_y$ is the moment of the cutting head in the Y direction, in N·mm; $M_z$ is the moment of the cutting head in the Z direction, in N·mm; $\theta$ is the position angle of the pick, in °; $\alpha$ is the friction angle between blade and rock, in °; $d_i$ is the distance between the $i$-th section and the centroid, in mm; $D$ is the cutting head diameter at the location of the cutting pick, in mm.

In order to analyze the dynamic reliability of the roadheader, when analyzing the load of the cutting head under complex conditions, the rock samples to be excavated can be tested to obtain their physical and mechanical properties. Combined with the force analysis of the cutting pick, the cutting head load can be calculated. The calculation process is shown in Fig 4.

**2.1.2 Motion analysis of the Swing process.** The horizontal lateral swing of the cutting part is achieved through the extension and retraction of the rotary hydraulic cylinder [18]. One end of the hydraulic cylinder is pivotally connected to the main body's support frame, while the other end is hinged to the gyration platform, enabling the entire cantilever to pivot with the gyration platform. Symmetrically arranged hydraulic cylinders on the gyration platform facilitate extension and retraction, causing the gyration platform to rotate along its axis. Specifically, when one hydraulic cylinder extends, the corresponding hydraulic cylinder on the opposite side contracts, inducing the yaw motion of the cutting section. Fig 5 illustrates a schematic diagram of the yaw process of the cutting section.

$AB$ is the length of the hydraulic cylinder when the cutting part is in the middle position; $A'B$ is the length of the hydraulic cylinder at its maximum extension when the cutting head moves to $L/2$; $OC$, $OC_1$ are the lengths of the cut parts; $\alpha$ is the angle at which the cutting part rotates when reaching $L/2$, $\alpha = L/2 \cdot OC'$; $\beta$ is the angle between the connecting line between the rotary center and the hinge center and the hydraulic cylinder when the cutting part is in the

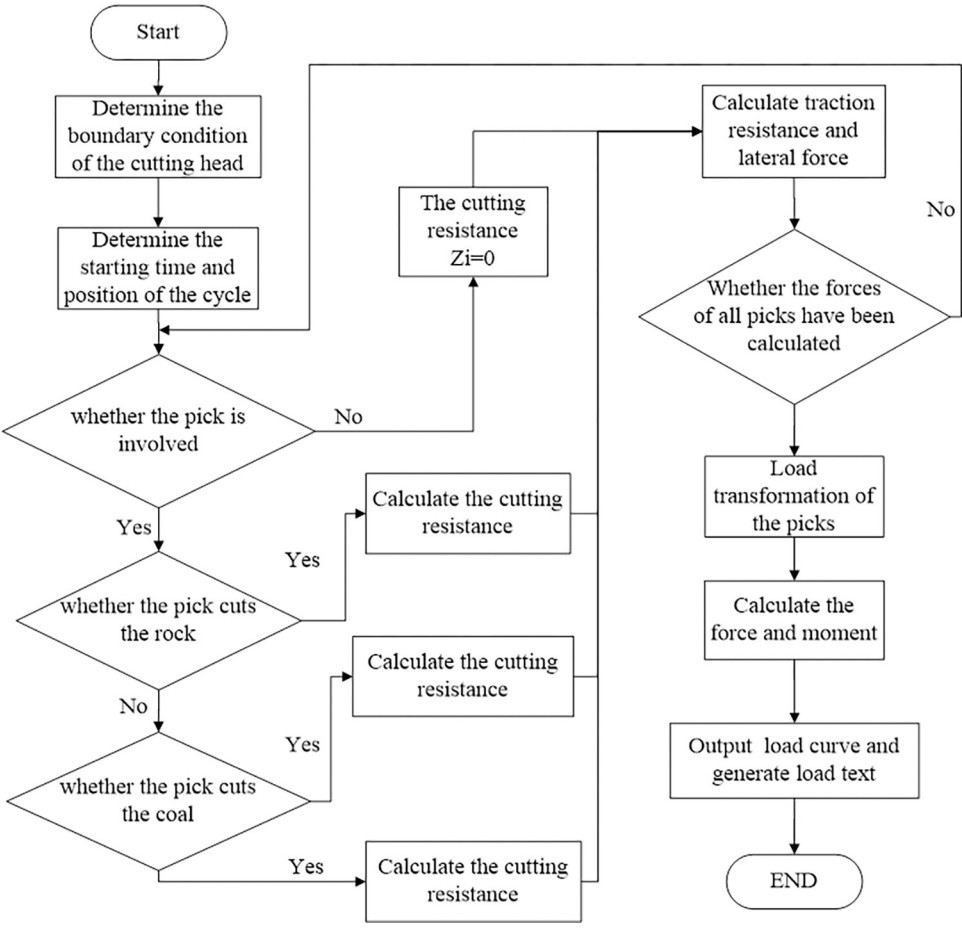

**Fig 4. The flow of cutting performance calculation.**

middle position; γ is the angle between the connection of the center point of the pivot and the connection between the center of the pivot and the center of the pivot. Since $OA = OA'$, $\gamma = 180 - \alpha/2$. The elongation of hydraulic cylinder is $Exl = A'B - AB$.

In the triangle $\triangle OAA'$:

$$\cos \alpha = \frac{OA^2 + OA'^2 - AA'^2}{2OA \cdot OA'} \tag{13}$$

$$AA' = \sqrt{OA^2 + OA'^2 - 2OA \cdot OA' \cdot \cos \alpha} \tag{14}$$

In the triangle $\triangle AA'B$:

$$\cos(\beta + \gamma) = \frac{AA'^2 + AB^2 - A'B^2}{2AA' \cdot AB} \tag{15}$$

$$A'B = \sqrt{AA'^2 + AB^2 - 2AA' \cdot AB \cdot \cos(\beta + \gamma)} \tag{16}$$

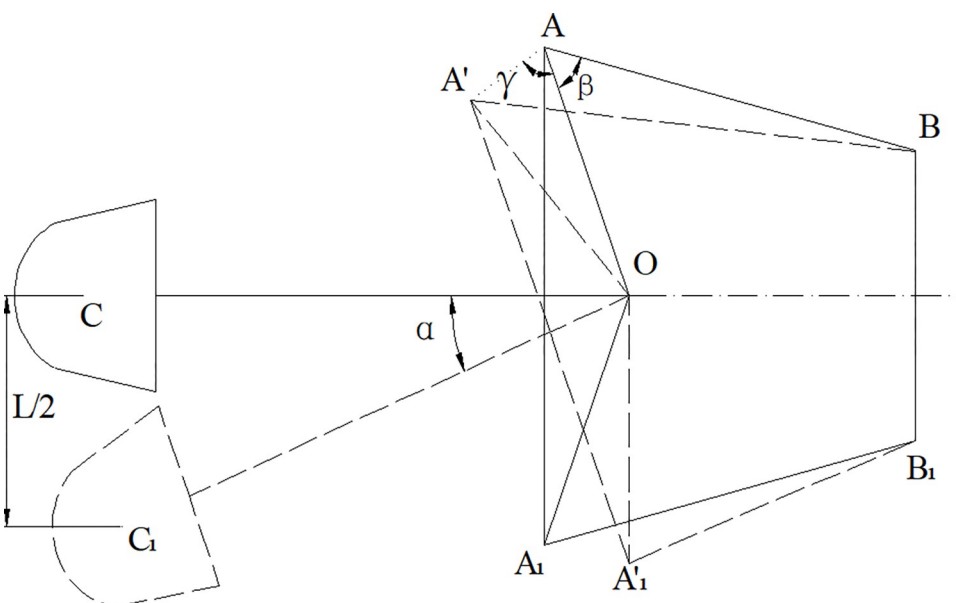

**Fig 5. Swing geometric model of roadheader.**

So, the elongation of the hydraulic cylinder can be calculated according to the following formula:

$$Exl = \sqrt{AA'^2 + AB^2 - 2AA' \cdot AB \cdot \cos(\beta + \gamma)} - AB \tag{17}$$

According to the structure and actual working conditions of this type roadheader, $AA = 332.9$mm, $AB = 646.7$mm, $OC = 3000$mm, $L = 1500$mm, $\alpha = 30°$, $\beta = 58.3°$, $\gamma = 54.6°$. The elongation of the hydraulic cylinder was calculated to be 187.9mm.

Based on the operational efficiency stipulated for the roadheader, the cutting time required for the yaw cutting can be computed. Subsequently, the extension speed of the hydraulic cylinder is determined by:

$$v_1 = \frac{Exl}{t} \tag{18}$$

During swing cutting operations of the roadheader, the aforementioned formula establishes the correlation between the extension speed of the hydraulic cylinder and the section width. This relationship becomes a crucial hydraulic cylinder driving parameter that requires configuration in subsequent analyses.

**2.1.3 Calculation and experimental verification of cutting head load.** The operational parameters were determined based on the geological conditions specific to this model's mining face. The properties of rock samples were determined, and the results of the property measurements are presented in Table 1. When calculating the load, the average of two measurement results was used for the computation. The cutting motor power is 75 kW, and the rotation

**Table 1. Property parameters of rock samples.**

| NO. | Density (kg/m³) | Elastic modulus (MPa) | Poisson's ratio | Tensile strength (MPa) | Compressive strength (MPa) | Firmness coefficient |
|---|---|---|---|---|---|---|
| 1 | 2610 | 18300 | 0.22 | 5.24 | 52 | 6.1 |
| 2 | 2620 | 17500 | 0.21 | 5.11 | 49 | 5.9 |

**Table 2. Relevant parameters of cutting head and pick.**

| | Parameter | Value |
|---|---|---|
| Cutting head | Swing speed (r/min) | 53 |
| | Maximum swing speed (m/min) | 1.1 |
| | Distance from cutting head to rotary center (mm) | 2408 |
| Cutting pick | Cutting angle of pick (°) | 45 |
| | Alloy head diameter (mm) | 15 |
| | Number of cutting picks involved | 34 |

speed of the cutting head is 53 r/min. Detailed parameters of the cutting head are provided in Table 2. Operating at full load, the maximum transverse swing speed of the cutting head is approximately 1.1 m/min. At this time, the expansion and contraction speed of the hydraulic cylinder is 2.2mm/s. The instantaneous load of the cutting head can be computed for two working conditions: transverse swing from left to right and transverse swing from right to left. The resulting cutting head load curves are illustrated in Fig 6.

Figs 7 and 8 visually depict a coherent relationship between the torque applied to the cutting head and the power consumption of the cutting head. As the roadheader's swing speed escalates, the absolute value of the cutting torque exhibits a continuous rise. Simultaneously, the power consumption of the cutting head also experiences an increase, as elucidated in Fig 9. It is noteworthy that, given the cutting power specification for this roadheader at 75 kW, it is imperative to maintain the swing speed within 1.1 m/min during the cutting process to ensure that the machine is not overloaded.

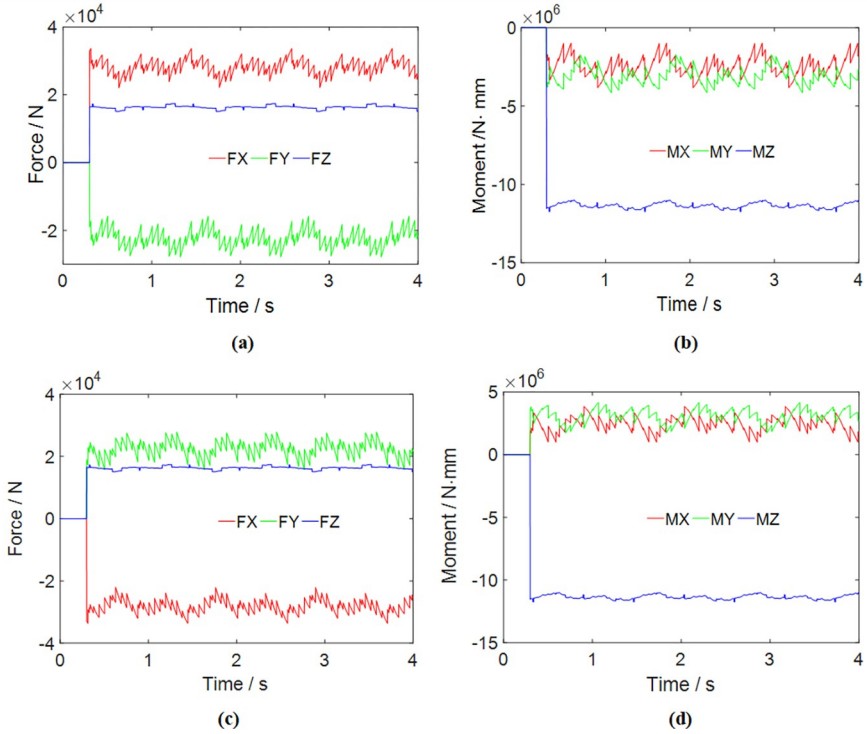

**Fig 6.** Load on the cutting head under different swing directions: (a) The cutting force when swing from left to right. (b) The cutting moment when swing from left to right; (c) The cutting force when swing from right to left; (d) The cutting moment when swing from right to left.

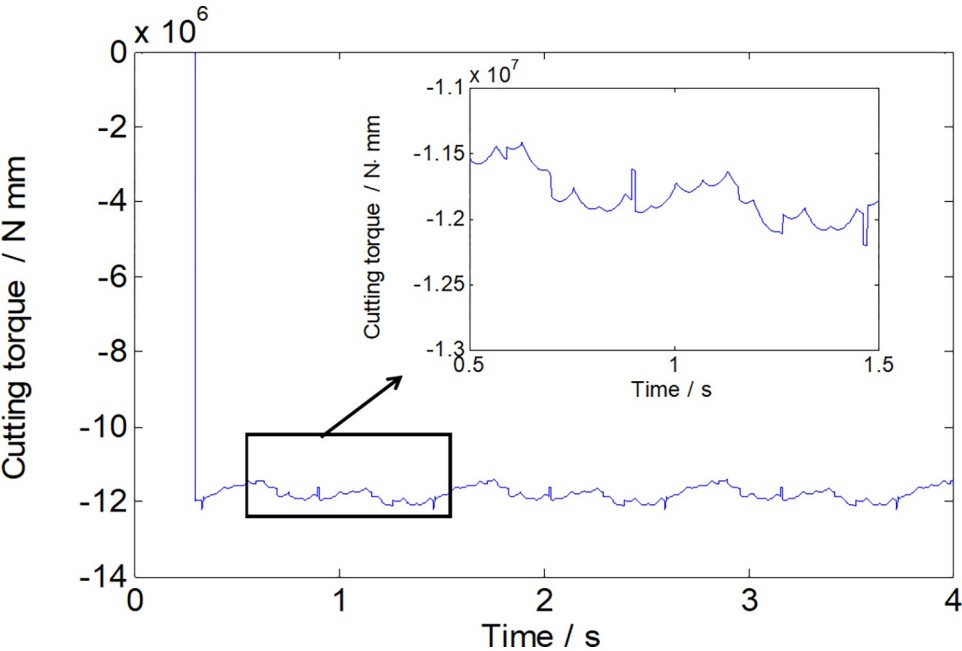

**Fig 7. The torque of the cutting head with a swinging speed of 1.1m/s.**

To validate the accuracy of the load calculations, a simulated working face mirroring the properties of the rock cut by the specified roadheader was created using cement, water reducing agent, and lime. The cutting part structure of the roadheader was replicated for cutting tests on this simulated working face, as illustrated in Fig 10. The torque during the cutting process was obtained by installing a torque sensor on the cutting motor torque shaft, and the torque value was obtained using a data acquisition device. The results of tests and simulations

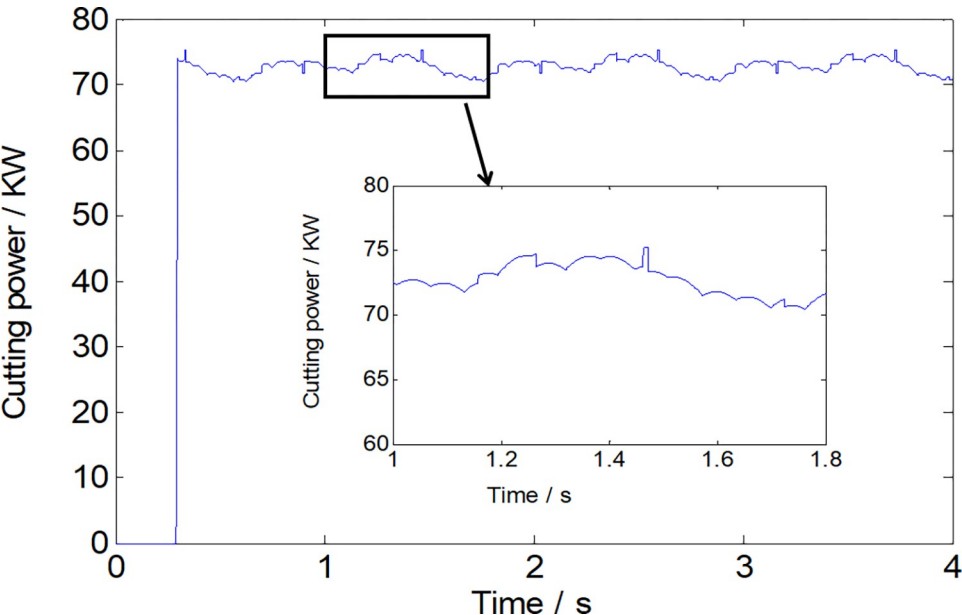

**Fig 8. The power consumption of the cutting head with a swinging speed of 1.1m/s.**

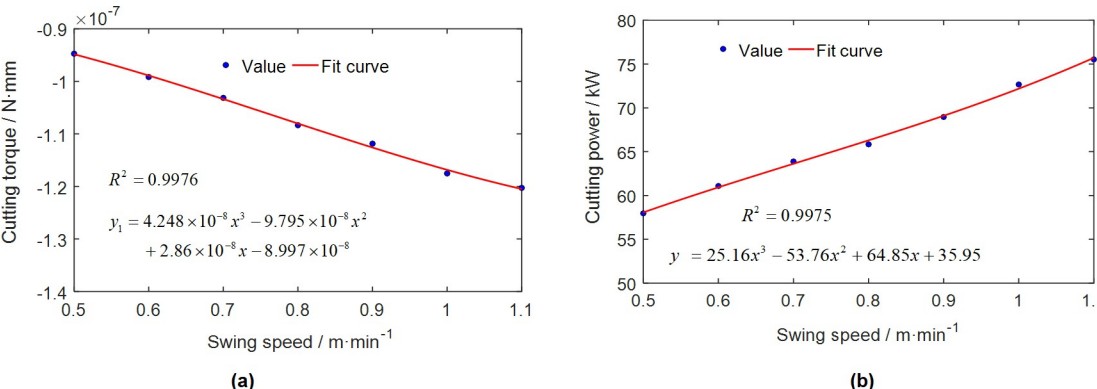

**Fig 9.** The torque and power with different swing speeds: (a)The torque with different swing speeds; (b) The power with different swing speeds.

comparing torque with varying swing speeds are presented in Fig 11 and Table 3. The maximum relative error between the effective torque values obtained through experimental and simulation methods under different conditions is 9.48%. This indicates the reliability of the load data derived from the theoretical calculations. Importantly, the calculated load data serves as a robust source of external excitation for multi-body dynamics simulations.

## 2.2 Establishment of multibody model

A 3D model of the cutting part was established and imported into ADAMS software [19]. Constraints were added reasonably based on the topological structure between various parts to achieve the normal operation of each action of the roadheader. The roadheader will be subjected to nonlinear impact loads during the process of cutting rocks, which will have adverse effects on the reliability of the key mechanical structure of the roadheader. Nonlinear impact load is the result of the cutting teeth on the cutting head breaking the rock, so studying the impact load on the cutting head is a prerequisite for studying the reliability and fatigue life of the mechanical structure of the roadheader. For a roadheader, the gyration platform is an

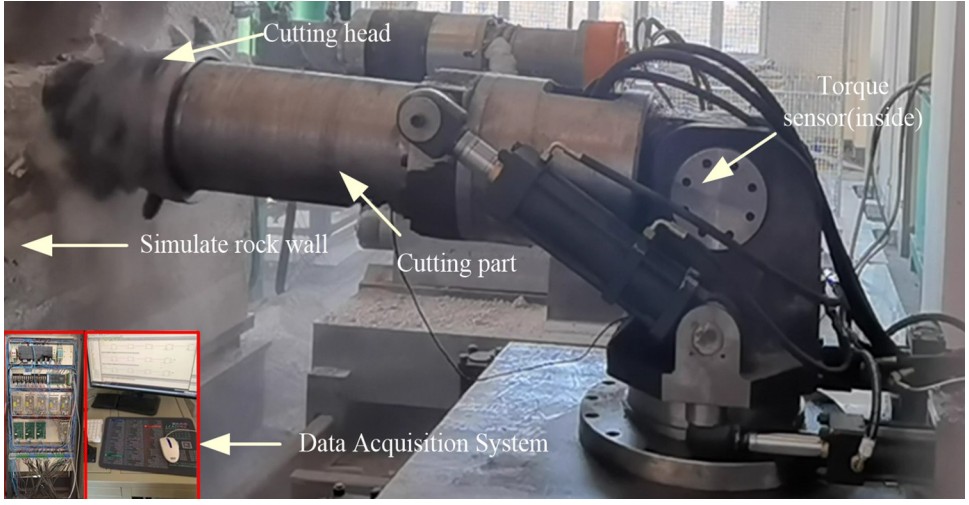

**Fig 10. Cutting test.**

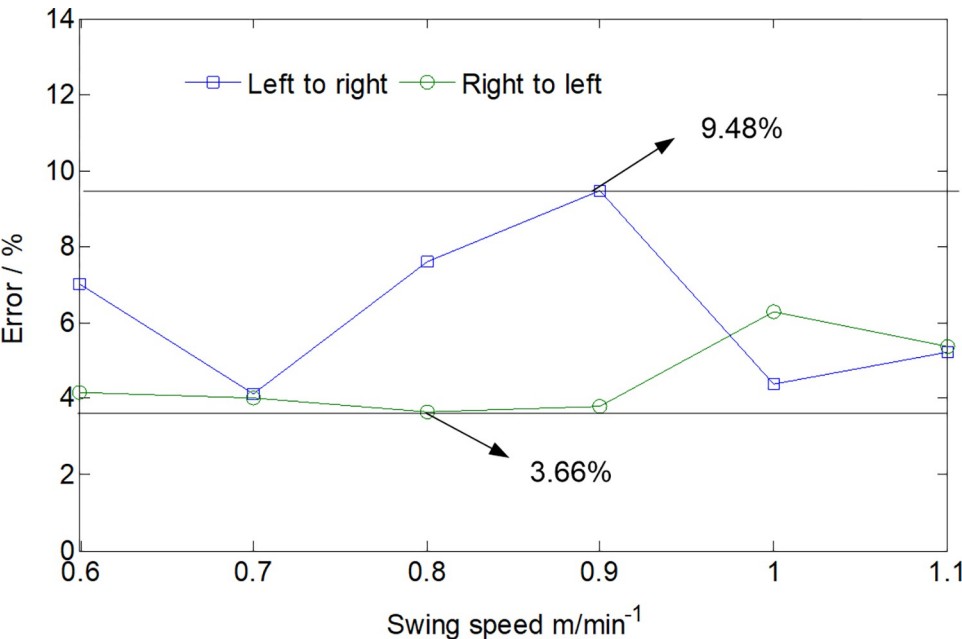

**Fig 11. The error between test and simulation results.**

important component that connects the main frame and supports the cutting arm. During the cutting process, the gyration platform needs to achieve the lifting and rotation of the cutting arm and withstand strong alternating loads, which is a weak link. If the gyration platform is damaged, it will affect the normal progress of excavation work. To scrutinize the dynamic characteristics of the gyration platform, finite element software was employed to discretize and model the gyration platform, generating a flexible body. The original model's rigid model was substituted with a flexible gyration platform [20,21]. Accounting for the interplay between rigid and flexible components and considering the distinctive attributes of the flexible platform, additional constraints and drives were introduced at the external connection points, and the established rigid-flexible coupling model is shown in Fig 12.

## 3. Results and discussion

To conduct the dynamic simulation, simulated loads were applied to the centroid of the cutting head, and the resulting dynamic behavior was analyzed. Fig 13 illustrates the variation in the three-dimensional acceleration of the cutting head during the roadheader's left-to-right cutting motion. Throughout the cutting process, the impact of nonlinear alternating loads induces the most severe vibration in the cutting head's vertical direction. The intensity of vibration is observed in the following order: X direction (vertical direction) > Y direction (swing direction) > Z direction (perpendicular to the working face direction). The effective

**Table 3. Torque values obtained through test and simulation.**

| Torque ×10⁷ N·mm | Swing speed from left to right (m/ min⁻¹) | | | | | | Swing speed from right to left (m/ min⁻¹) | | | | | |
|---|---|---|---|---|---|---|---|---|---|---|---|---|
| | **0.6** | **0.7** | **0.8** | **0.9** | **1.0** | **1.1** | **0.6** | **0.7** | **0.8** | **0.9** | **1.0** | **1.1** |
| Test | 0.9215 | 0.9854 | 0.9977 | 1.0014 | 1.1212 | 1.1394 | 0.8341 | 0.8598 | 0.9045 | 0.9342 | 0.9741 | 0.9987 |
| Simulation | 0.9912 | 1.0278 | 1.0798 | 1.1063 | 1.1725 | 1.2021 | 0.8703 | 0.8958 | 0.9389 | 0.9712 | 1.0395 | 1.0553 |
| Error /% | 7.03 | 4.13 | 7.6 | 9.48 | 4.38 | 5.22 | 4.16 | 4.02 | 3.66 | 3.81 | 6.29 | 5.36 |

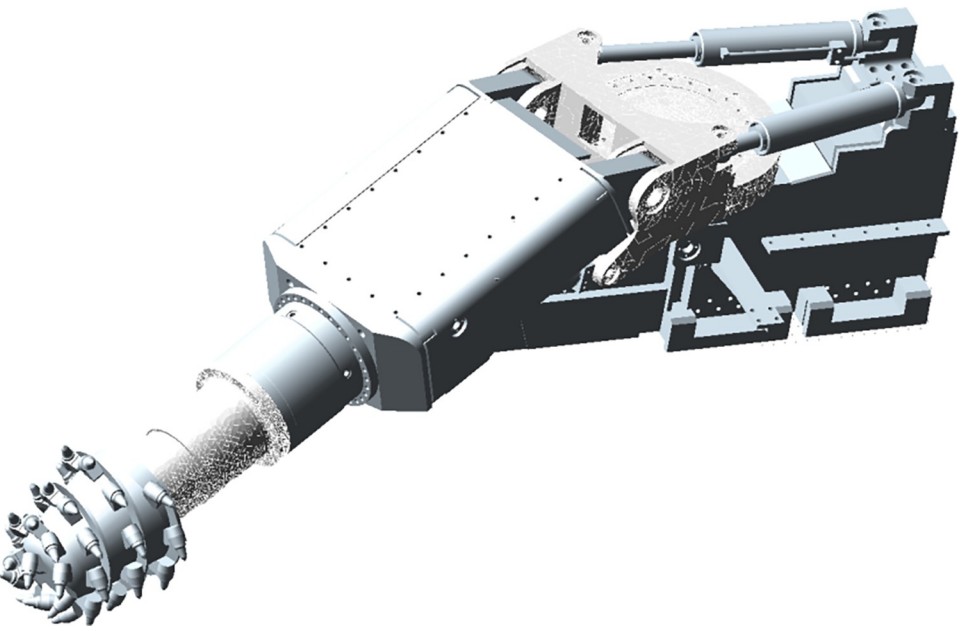

**Fig 12. Rigid-flexible coupling mode.**

acceleration values (root mean square, RMS) of the cutting head in the X, Y, and Z directions after achieving stable cutting are 34.340 m/s$^2$, 16.490 m/s$^2$, and 12.476 m/s$^2$, respectively.

According to the topological relationship of the cutting part structure, multiple external connection points were established for each connecting ear of the gyration platform to connect with other structures, as depicted in Fig 14. Fig 15 shows the vibration acceleration of Node 62796 in three directions. Due to its connection to the height adjustment cylinder and the lateral load transmitted by the cutting part during the swing process, the vibration is most pronounced in the Y direction during cutting. Table 4 provides the vibration acceleration data for

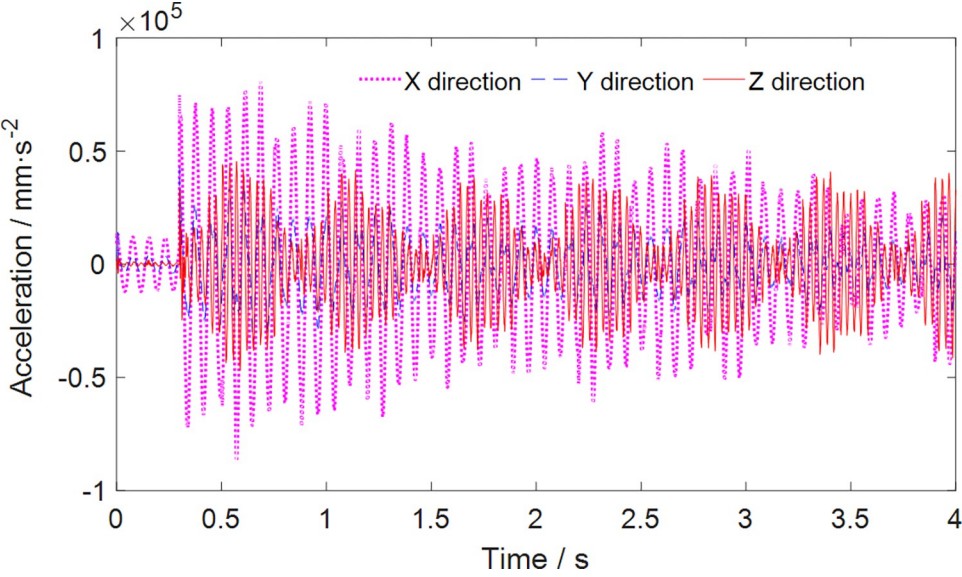

**Fig 13. Vibration acceleration in three directions of the cutting head.**

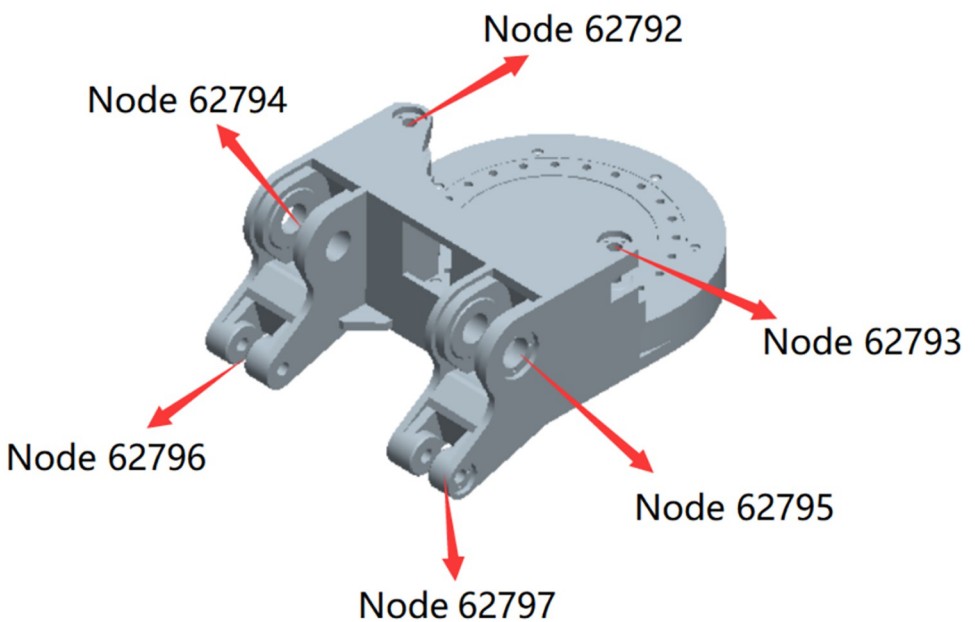

**Fig 14. Location of the main connection points of the gyration platform.**

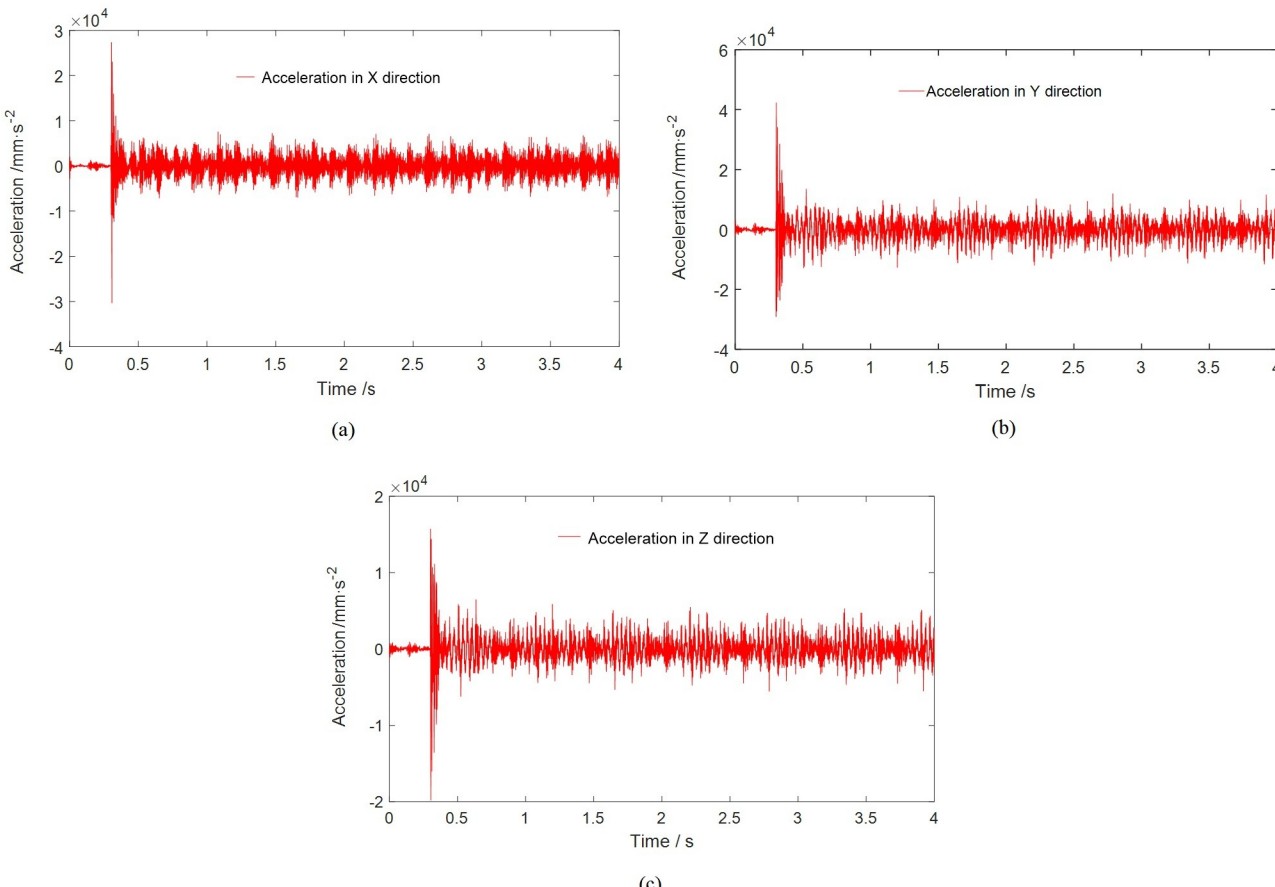

**Fig 15.** Vibration acceleration of Node 62796: (a) Acceleration in X direction; (b) Acceleration in Y direction; (c) Acceleration in Z direction.

**Table 4. Acceleration values of the main connection points.**

| Node | Acceleration in X direction (mm/s⁻²) | | | Acceleration in Y direction (mm/s⁻²) | | | Acceleration in Z direction (mm/s⁻²) | | |
|---|---|---|---|---|---|---|---|---|---|
| | Max | Min | RMS | Max | Min | RMS | Max | Min | RMS |
| 62792 | 5910.76 | -5063.30 | 795.73 | 14609.84 | -21817.43 | 1969.08 | 15856.64 | -10845.16 | 1392.49 |
| 62793 | 13972.77 | -7586.25 | 1129.63 | 20373.89 | -72269.58 | 1828.56 | 16870.99 | -29605.91 | 1231.56 |
| 62794 | 7375.84 | -8094.05 | 676.19 | 18980.33 | -11822.21 | 1798.15 | 13418.48 | -9260.67 | 1156.97 |
| 62795 | 7586.76 | -7091.91 | 1191.58 | 14786.01 | -22936.24 | 1950.52 | 6071.20 | -12462.99 | 1308.61 |
| 62796 | 27335.31 | -30331.96 | 2555.64 | 42313.56 | -29097.02 | 3845.51 | 15732.01 | -19798.73 | 1773.83 |
| 62797 | 17636.81 | -25098.91 | 2367.93 | 15753.24 | -21661.75 | 3716.79 | 8693.16 | -10851.31 | 2128.44 |

the pertinent nodes of the gyration platform in three directions. The vibration patterns of Node 62797 and Node 62796 at the connecting ear positions beneath the gyration platform exhibit a consistent order: Y direction (swing direction) > X direction (vertical direction) > Z direction (perpendicular to the working face direction). For Node 62764 and Node 62795 at the connecting ear positions, as well as Node 62792 and Node 62793 at the connecting ear positions situated behind the gyration platform, the hierarchy of vibration levels follows the sequence Y > Z > X.

By utilizing ADAMS/Vibration for modal analysis of the roadheader model, both the road-header mode and its corresponding vibration shapes can be determined. This analysis allows for the calculation of modal natural frequencies and characteristic values for each mode, as presented in Fig 16 and Table 5. The frequencies of the 1st to 2nd order modes are relatively small, and the corresponding vibration shapes are the upper and lower torsional deformation of the ears connecting the gyration platform and the height adjusting cylinder around the center of the gyration platform. The vibration shapes of the 3rd to 4th order modes and the 6th to 8th order modes are basically consistent with the vibration trend of the 1st to 2nd order modes. However, there are large deformations at the connection between the gyration platform and the rotary cylinder, the connecting rib plate in the middle of the left and right ears and the left and right sides of the rotary plate. In the 5th mode shape, the maximum deformation position of the gyration platform is concentrated in the ear part of the side connected with the height adjusting cylinder, and there is a certain bending at the tail of the rotary plate.

Fig 17 illustrates the deformation trend of the gyration platform at the moment of maximum stress during a left-to-right swing, serving as an example. The illustration unveils a

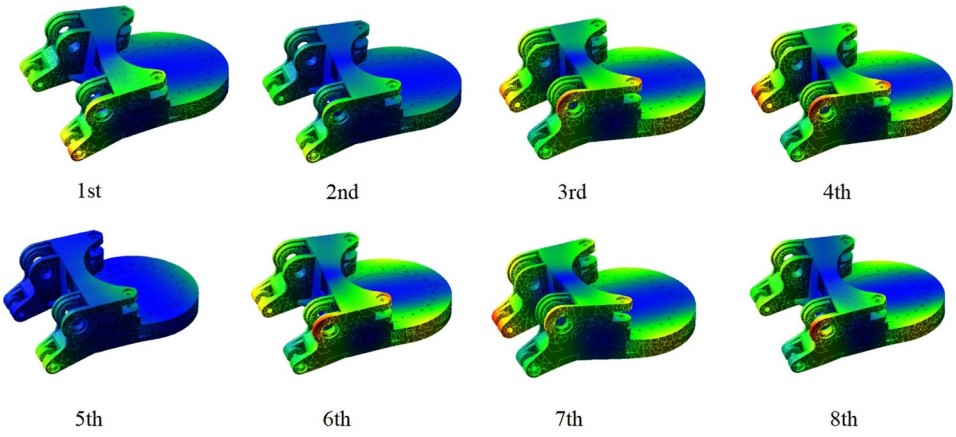

**Fig 16. Modal shape of the gyration platform.**

**Table 5. Modal frequency of the gyration platform.**

| Modal order | 1st | 2nd | 3rd | 4th | 5th | 6th | 7th | 8th |
|---|---|---|---|---|---|---|---|---|
| Frequency (Hz) | 1.3207 | 2.6256 | 87.766 | 88.273 | 113.366 | 114.549 | 116.099 | 292.813 |

distinct "convex" deformation pattern observed in the gyration platform throughout the cutting process, and the deformation predominantly manifests in the ear sections of the lifting cylinder and the rear portion of the gyration platform.

Fig 18 illustrates the deformation curve of node 33, representing the point of maximum stress on the gyration platform. At 0.325 seconds, the deformation at Node 33 reaches its peak value of 0.117mm. Moreover, the deformation of the gyration platform closely follows the changes in applied force, reaching maximum deformation at the loading moment and gradually stabilizing thereafter. Ultimately, the deformation magnitude fluctuates around 0.06 mm.

Fig 19 presents the maximum deformation of the gyration platform during left-to-right swing cutting with different speeds. The gyration platform's deformation exhibits a parallel trend to its stress variation, wherein higher swing speeds result in more pronounced deformation.

Fig 20 presents the stress distribution of the gyration platform in the two lateral swinging directions. The gyration platform experiences a maximum stress of 258.71 MPa during right-to-left lateral swinging cutting, while during left-to-right lateral swinging cutting, the maximum stress escalates to 364.57 MPa. When cutting rock from left to right, the cutting resistance force on the cutting picks aligns with the gravitational direction of the cutting section, encompassing the cutting head. In contrast, during cutting from right to left, the cutting resistance force acts in an upward direction, partially mitigating the impact of gravitational force on the stress exerted on the gyration platform. Consequently, cutting from right to left enhances the dynamic characteristics of the gyration platform. In both operational scenarios, the regions of elevated stress on the gyration platform are chiefly concentrated around the bolt holes connecting it to the slewing bearing.

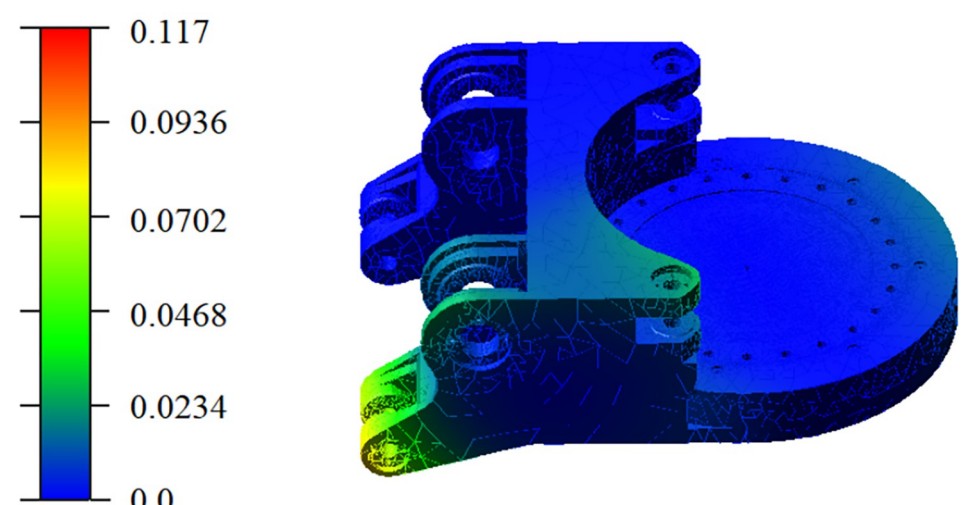

**Fig 17. Deformation trend of the gyration platform.**

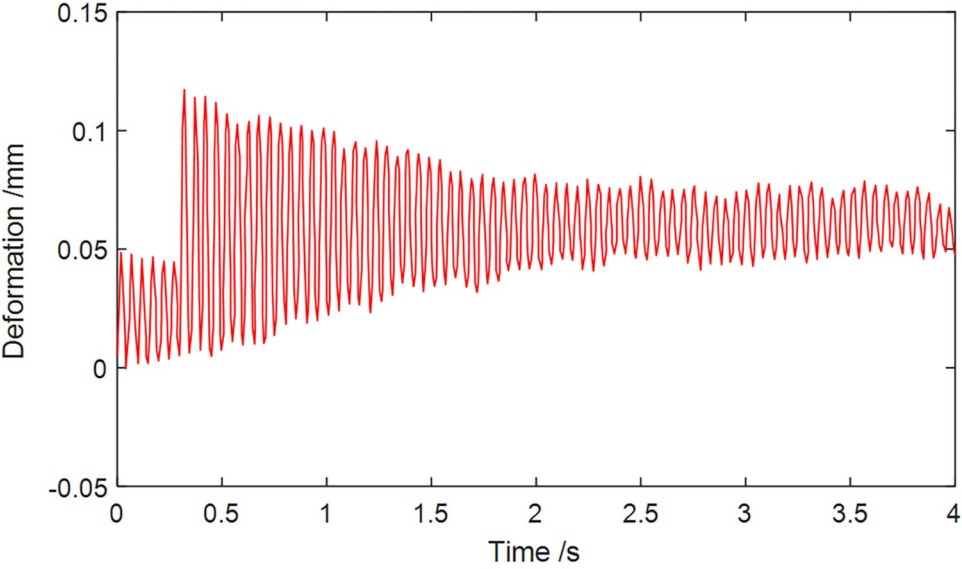

**Fig 18. Deformation variation of Node 33.**

When the roadheader swings from left to right, the evolution of maximum stress at Node 33 on the gyration platform is depicted in Fig 21. The graph illustrates that during the unladen phase from 0 to 0.3 seconds, the gyration platform undergoes a relatively stable stress state attributed to the combined effects of the cantilever and its own gravitational force. At 0.3 seconds, as the roadheader initiates the swinging and cutting motion, the instantaneous loading of the cutting head swiftly elevates the force exerted on the gyration platform. Following the establishment of stable cutting conditions, the force acting on the gyration platform gradually stabilizes, exhibiting periodic fluctuations.

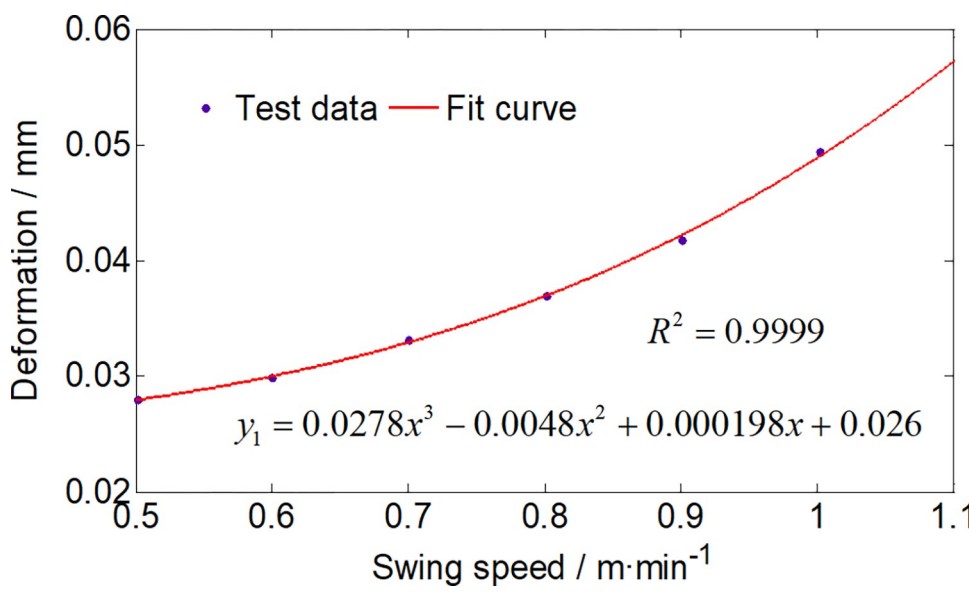

**Fig 19. Deformation of the gyration platform with different swing speeds.**

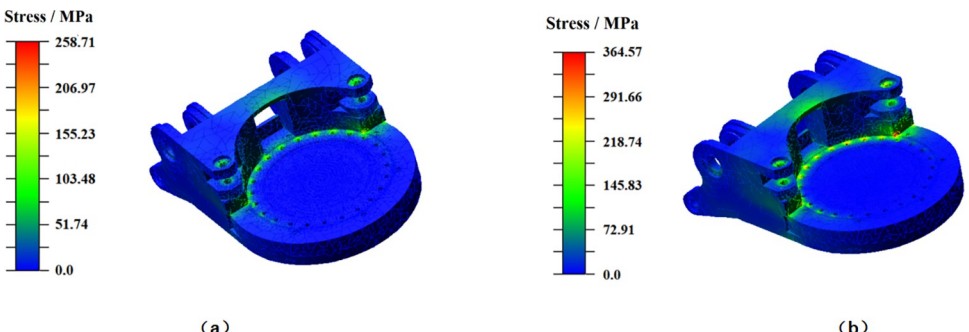

**Fig 20. Stress distribution of gyration platform.** (a) Swing from right to left. (b) Swing from left to right.

The graphical representation in Fig 22 illustrates the variation in maximum stress experienced by the gyration platform under diverse swing speeds and directional conditions. At consistent swing speeds, the gyration platform undergoes elevated maximum stress levels during a left-to-right swing in comparison to a right-to-left swing. The analysis reveals that as the swing speed increases, there is a discernible augmentation in the magnitude of maximum stress exerted on the gyration platform.

According to the Miner's Rule, the load acting on the gyration platform is composed of $m$ different stresses [22]. The number of cycles at each stress level is denoted as $n_1, n_2, \ldots, n_m$, and the corresponding fatigue life is represented as $N_1, N_2, \ldots, N_m$. Following the linear accumulation principle, the total damage to the rotary platform is calculated according to Formula (19):

$$R_s = D_s = \sum_{i=1}^{m} \frac{n_i}{N_i} \tag{19}$$

When $D$s = 1, the distortion energy density of the gyration platform reaches its limit and fails.

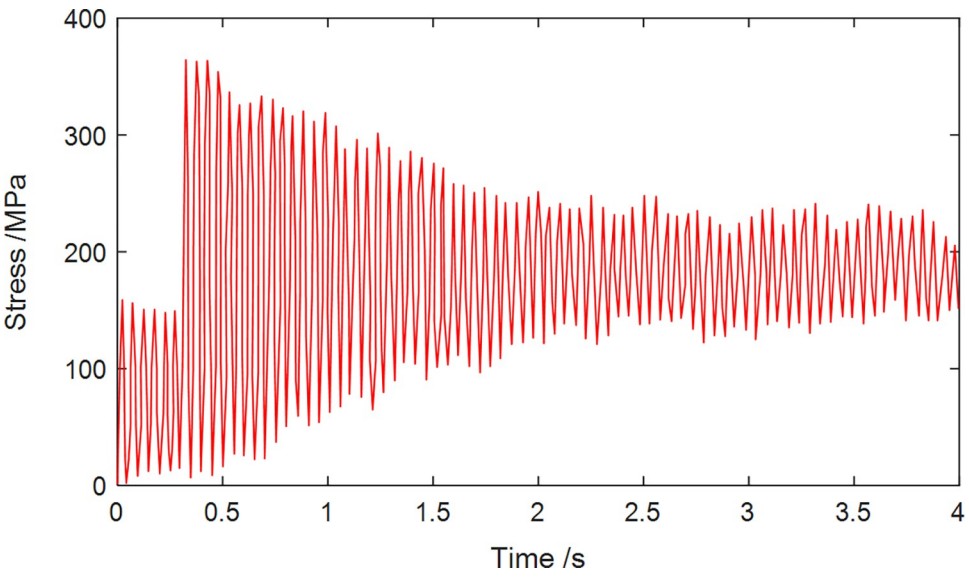

**Fig 21. The stress variation of Node 33.**

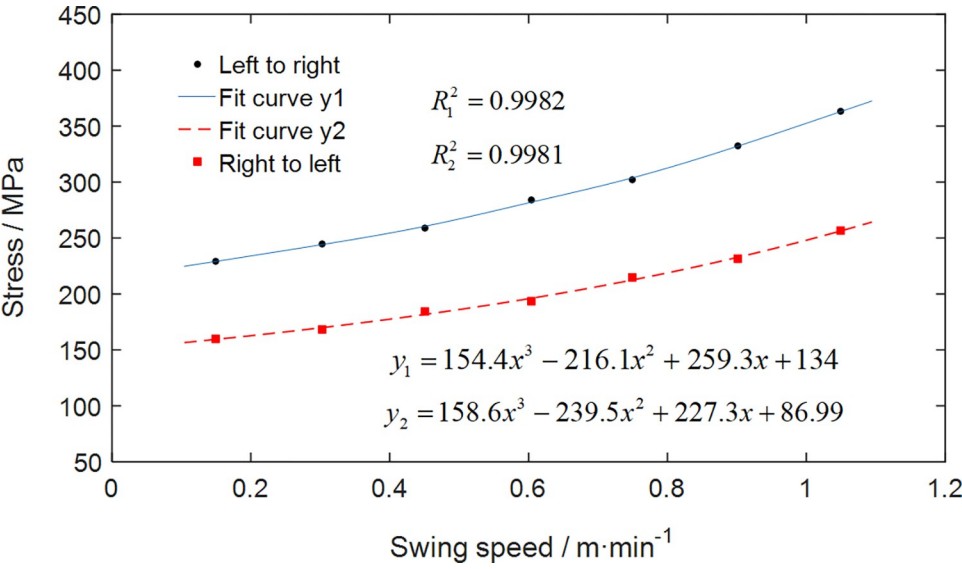

**Fig 22. Maximum stress of the gyration platform with different swing speeds.**

In the execution of a thorough fatigue analysis, the time-history load file, stress results file, and modal file derived from the dynamic simulation are systematically extracted and subsequently imported into the Nsoft software [23]. The material properties of Q235A are employed for characterization of the gyration platform material, facilitating the establishment of the corresponding S-N curve. A comprehensive life analysis was conducted on the gyration platform, leading to the identification of 10 nodes that manifest notable fatigue damage during the left-to-right swing, as shown in Table 6.

The results file, encompassing critical information regarding fatigue life and damage, was imported into ADAMS to derive the fatigue life distribution of the gyration platform, as depicted in Fig 23. Analysis of Table 6 and Fig 23 indicates that nodes exhibiting the shortest fatigue life are primarily situated in the frontal region of the gyration platform, in proximity to the bolt hole edges connecting the gyration platform and the rotary bearings. Among these nodes, node 33 demonstrates the lowest fatigue life, enduring a maximum cyclic load of $5.366 \times 10^6$ cycles.

**Table 6. Fatigue damage analysis results of the gyration platform.**

| No. | Node | Fatigue life ($10^6 \cdot$s) | Fatigue damage ($10^{-6} \cdot$mm) |
|---|---|---|---|
| 1 | 33 | 0.537 | 5.954 |
| 2 | 76 | 0.691 | 3.746 |
| 3 | 36 | 1.101 | 0.337 |
| 4 | 3793 | 1.365 | 0.242 |
| 5 | 1420 | 5.059 | 0.126 |
| 6 | 1426 | 8.135 | 0.116 |
| 7 | 1586 | 12.548 | 0.101 |
| 8 | 1429 | 19.886 | 0.058 |
| 9 | 1594 | 21.349 | 0.047 |
| 10 | 78 | 24.342 | 0.026 |

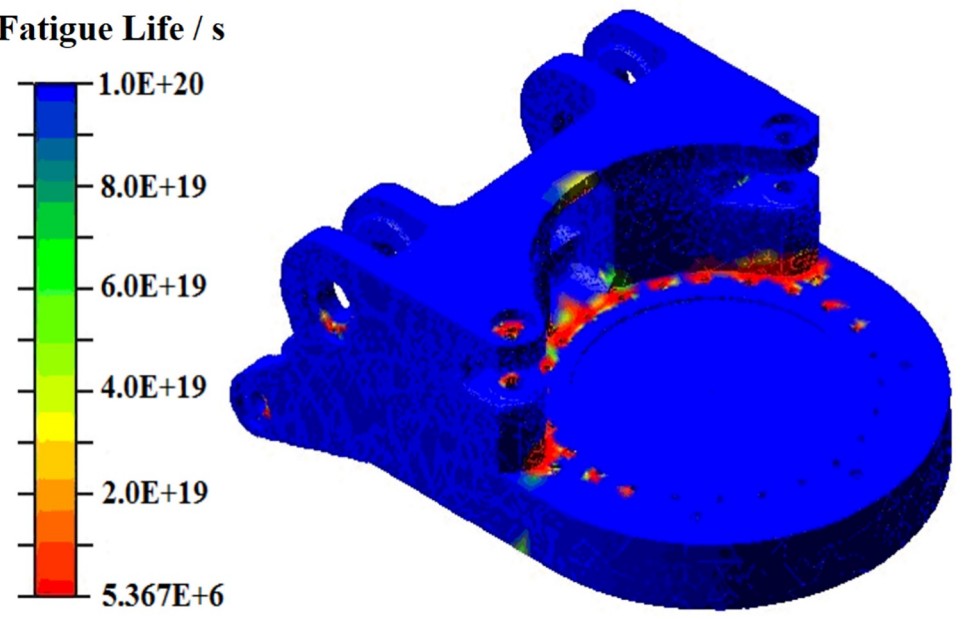

**Fig 23. Fatigue life distribution of gyration platform.**

Fig 24 delineates the fluctuation in the minimum fatigue life across nodes within the gyration platform during left-to-right swing cutting at varying speeds. Noteworthy is the discernible augmentation in dynamic stresses experienced by the gyration platform with an escalation in swing speed, thereby manifesting a precipitous decline in its fatigue life. Consequently, judicious selection of an optimal swing speed, contingent upon the hardness of the rock during swing cutting, becomes imperative to uphold the reliability of the gyration platform. When

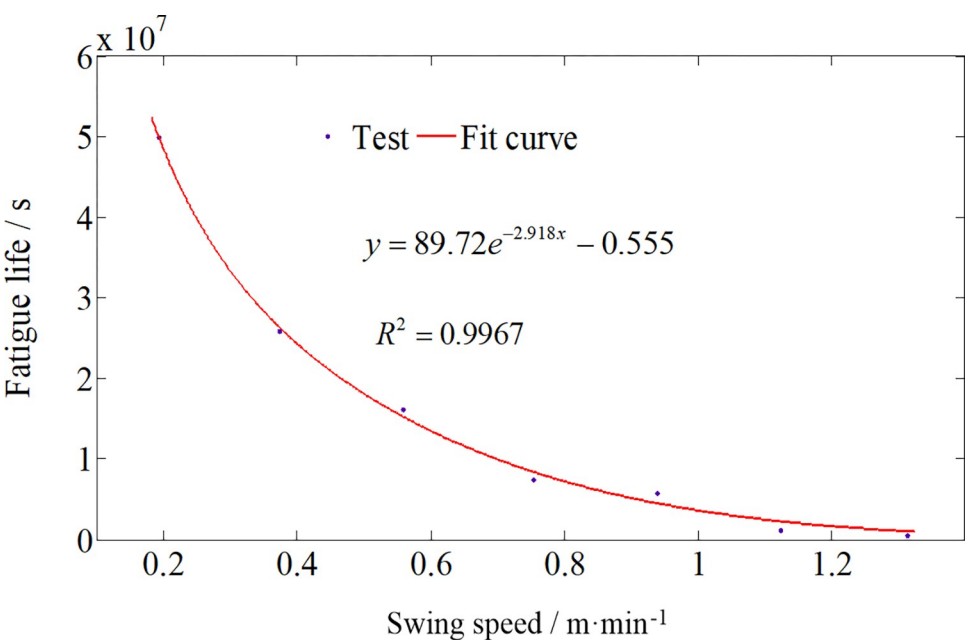

**Fig 24. Relationship between fatigue life and swing speed of gyration platform.**

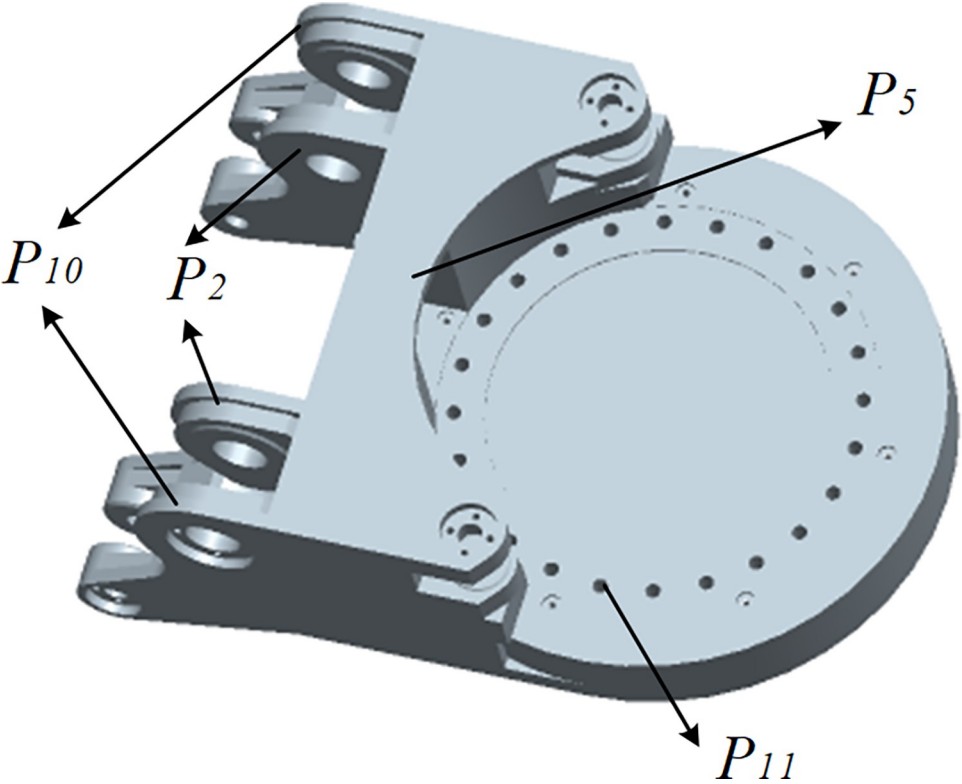

**Fig 25. Optimization parameters of gyration platform.**

confronted with the challenge of cutting hard rock, exercising caution is imperative to ensure the reliability of pivotal components. Conversely, when faced with the cutting of soft rock, the selection of an appropriate swing speed can be based on the cutting power. To safeguard the dependable operation of the roadheader, a judicious approach is advocated. Specifically, a relatively slower swing speed is recommended during left-to-right cutting, whereas a comparatively faster swing speed may be chosen for right-to-left cutting.

The locations with high stress on the gyration platform are distributed at the bolt holes connected to the rotary bearings at the front of the gyration platform, the middle part of the connecting rib plate of the ear connected to the rotary hydraulic cylinder, the ear connected to the lifting hydraulic cylinder, and the ear connected to the cutting motor box. As shown in Fig 23, these locations are also areas with lower fatigue life. In order to reduce the maximum stress of the gyration platform during the cutting process and improve its fatigue life, the parameters P10 (thickness of the two outer rib plates), P2 (thickness of the two inner rib plates), P5 (thickness of the upper ear connecting rib plates), and P11 (diameter of the bolt hole connected to the rotary bearing) were optimized, as shown in Fig 25.

According to the design dimensions of the cutting machine case and the relative position between the body frame and the hydraulic cylinder, the variation range of parameters P10, P2, and P5 is 25-30mm, and the variation range of parameter P11 is 8.1–9.9mm. The optimization design objective parameters are to minimize the increase in mass of the gyration platform and reduce the maximum stress of the gyration platform. Fig 26 shows the optimized design nodes of different parameters obtained from simulation analysis.

20 sampling points for optimized design were determined based on the numerical range of design parameters, and the MOGA method was selected for optimization based on model

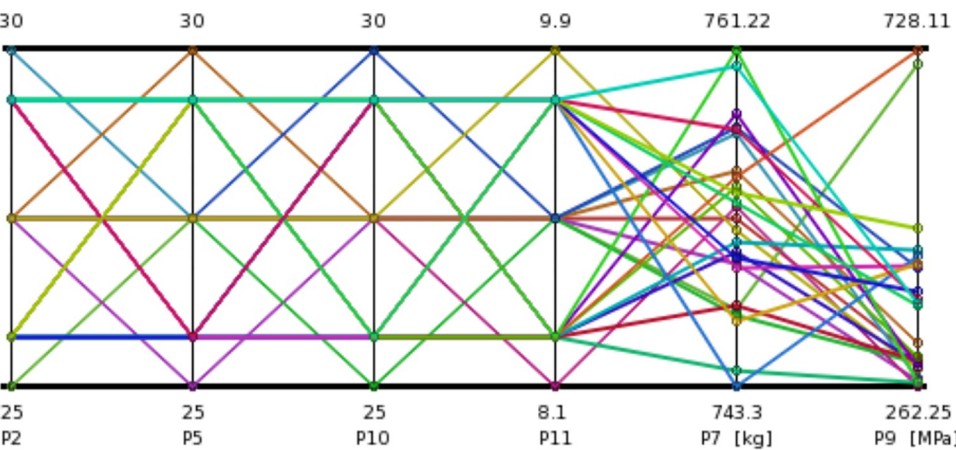

**Fig 26. Optimized design nodes of different parameters.**

characteristics. The simulation provides three optimization schemes, as shown in Table 7, and by comparing the relevant parameters in different schemes, scheme 1 was selected as the optimization scheme. The mass of the optimized model is 749.18 kg, representing a mere 9.03 kg increase compared to its pre-optimized state. Furthermore, the maximum stress in the optimized model measures 293.23 MPa, indicating a substantial reduction of 71.34 MPa from the pre-optimization stress level of 364.57 MPa. These analyses underscore a noteworthy enhancement in the stress condition of the optimized gyration platform.

## 4. Conclusions

Based on the analysis of individual cutting pick forces, a transformation of all cutting forces to the centroid of the cutting head was executed via coordinate transformation. This methodology effectively centralized the forces, facilitating a comprehensive analysis of the instantaneous loads acting on the cutting head under diverse lateral swing conditions. The resultant loads served as external excitations for the dynamic simulation of the cutting unit model.

Noteworthy insights were gleaned through dynamic simulation analysis of the rigid-flexible coupled multibody model of the roadheader's cutting part. During left-to-right swing cutting, the gyration platform experienced markedly elevated maximum stress in comparison to right-to-left swing cutting. This heightened stress was predominantly localized in the vicinity of the bolt holes interconnecting the gyration platform and the rotary bearings. The fatigue life analysis shows that the node with the shortest fatigue life is mainly located in the front area of the rotating platform, near the edge of the bolt hole connecting the rotating platform and the rotating bearing. With an escalation in swing speed, there was a conspicuous intensification in both the maximum stress and deformation of the gyration platform, consequently leading to a marked reduction in its fatigue life. It is noteworthy that, when confronted with the challenge

**Table 7. Results of different optimization schemes.**

| NO. | P2 (mm) | P5 (mm) | P10 (mm) | P11 (mm) | Mass (kg) | Maximum stress (MPa) |
|-----|---------|---------|----------|----------|-----------|----------------------|
| 1 | 27.332 | 26.666 | 26.532 | 9.0594 | 749.18 | 293.23 |
| 2 | 26.618 | 28.297 | 26.939 | 8.5446 | 755.62 | 295.51 |
| 3 | 27.163 | 27.291 | 27.124 | 8.7714 | 753.86 | 298.73 |

of cutting hard rock,it is necessary to reduce the swing speed to ensure the reliability of key components.

In order to improve the reliability of the gyration platform, the relevant structural parameters of the gyration platform, such the thickness of the two outer rib plates, the thickness of the two inner rib plates, the thickness of the upper ear connecting rib plates, and the diameter of the bolt hole connected to the rotary bearing, were optimized and designed. The analysis results indicate that the stress state of the optimized gyration platform can be significantly improved. The research findings not only furnish valuable guidance for enhancing the structural integrity of the gyration platform but also provide insights into the judicious selection of swing speeds, thereby contributing significantly to the overall enhancement of roadheader performance.

## Author Contributions

**Formal analysis:** Yang Ge.

**Funding acquisition:** Wei Liu.

**Investigation:** Shan Gao, Yang Ge.

**Methodology:** Shan Gao, Lijuan Zhao.

**Project administration:** Lijuan Zhao.

**Resources:** Zhen Tian, Lijuan Zhao.

**Supervision:** Wei Liu.

**Validation:** Zhen Tian.

**Visualization:** Quan Sun.

**Writing – original draft:** Shan Gao.

**Writing – review & editing:** Zhen Tian.

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
