## [Decision Letter · Decision Letter 0]

24 Nov 2023

PONE-D-23-35060Dynamic characteristics and fatigue life analysis of gyration platform of roadheaderPLOS ONE

Dear Dr. Tian,

Thank you for submitting your manuscript to PLOS ONE. After careful consideration, we feel that it has merit but does not fully meet PLOS ONE’s publication criteria as it currently stands. Therefore, we invite you to submit a revised version of the manuscript that addresses the points raised during the review process.

 Please, address all the comments made by the reviewers, and highlight all the changes in the revised manuscript. More explanations on the methods and tools selected to perform the study and on how the data were obtained are required. Please, also give more details on the numerical calculations of the cutting head load. Finally, please, explain how a rectangular groove was produced by a dome head.

We look forward to receiving your revised manuscript.

Kind regards,

Antonio Riveiro Rodríguez, PhD

Academic Editor

PLOS ONE

 [This work was supported in part by The Science and Technology Tackling Key Project of Henan Province under Grant 232102321097, and in part by Colleges and Universities Scientific Research Projects of Henan Province under Grant 21B440004 and in part by The Science and Technology Research Project of Zhoukou city under Grant 2022GG01008.].  

5. We note that Figure(s) 8, 10, 12, 15, 18 and 21 in your submission contain copyrighted images. All PLOS content is published under the Creative Commons Attribution License (CC BY 4.0), which means that the manuscript, images, and Supporting Information files will be freely available online, and any third party is permitted to access, download, copy, distribute, and use these materials in any way, even commercially, with proper attribution. For more information, see our copyright guidelines: http://journals.plos.org/plosone/s/licenses-and-copyright.

a. You may seek permission from the original copyright holder of Figure(s) 8, 10, 12, 15, 18 and 21 to publish the content specifically under the CC BY 4.0 license. 

6. Please upload a copy of Figure 13, to which you refer in your text on page 9. If the figure is no longer to be included as part of the submission please remove all reference to it within the text.

Reviewers' comments:

Reviewer's Responses to Questions

**Comments to the Author**

1. Is the manuscript technically sound, and do the data support the conclusions?

Reviewer #1: No

Reviewer #2: Partly

2. Has the statistical analysis been performed appropriately and rigorously? 

Reviewer #1: No

Reviewer #2: No

3. Have the authors made all data underlying the findings in their manuscript fully available?

Reviewer #1: No

Reviewer #2: Yes

4. Is the manuscript presented in an intelligible fashion and written in standard English?

Reviewer #1: Yes

Reviewer #2: No

5. Review Comments to the Author

Reviewer #1: Manuscript entitled ‘Dynamic characteristics and fatigue life analysis of gyration platform of roadheader’ analyzed the cutting parameters of a roadheader during the excavation process. Some dynamic data was captured and discussed regarding the position and orientation or the roadheader. The authors believed that this study could provide a reference for the improvement of the gyration platform structure and the selection of a suitable swing speed. This research cannot be accepted due to the following arguments.

1.The methods and the tools that the authors selected to carry out the research were too rough and simple. The authors are not familiar with the real working conditions of an excavator, there are many types for different working environments, to be straight forward and simple, coal cutter is coal cutter, roadheader is roadheader, one is for long wall mining face, one is for roadway excavation. The authors need to add more details of the differences, regarding the working conditions of them, rather than mix them together, sometimes declare that the roadheader is also adaptable for long wall mining face, which is clearly not precise.

2.As stated by the authors, ‘In order to ensure the reliability of the gyration platform, when cutting in different swing directions, the swing speed should be reasonably selected based on the hardness of the coal and rock’. How did the authors consider the up and down movement, rather than just left and right, the real scenario for a roadheader is all directions, rather than just horizontal movement. Though it is acceptable to just discuss the horizontal one, it is indeed necessary to also consider the vertical one, at least to add some discussions regarding the limitations of this study.

3.In section 2.1.3, the authors stated that the cutting head load was calculated by Matlab software, but the inner process is lacking, the purpose and the results from the calculation are also unknown, which both make this section rough and insufficient in details.

4.In section 2.1.3, the authors mentioned firmness coefficient, what was the strength values of the rock?

5.Also in section 2.1.3, how did the authors obtain the data? it is a necessity to briefly introduce the methods, or the arrangement of the sensors that captured all data.

6.The most serious flaw of this paper exists in Fig. 8, obviously the cutting test couldn't shape a rectangular groove as shown in the figure, the cutting head was an overall domed appearance, how did it possible for a domed head cut a rectangular groove? from this point, there is no doubt that the paper suffers flaw of factitious results from a scientific research. Therefore, from this section and onward, there is no need to read it any more, and it directly hinders a potential positive decision of this submission.

Reviewer #2: (1) The fatigue lifespan expression and/or result are not enough in the current manuscript that is not consist with the defined TITLE.

(2) The analysis reference is not referring the classical strength theory despite the manuscript filling with enough stress and strain maps.

(2) The current English expression need to improve and polish.

(3) The figures are low resolution without scope of academic publish standards.

(4) The fitting curves are low in physical recombination without enough factors illustration in engineering applications. Why it is high-order function with the abscissa? (Fig. 7 and Fig. 17)

(5) Figure 1 and Figure 2 should add the 3D model to support the agreed assumption before the raised question.

(6) The line diagrams in the referred figures should be normalized for a uniform style, including the font, size, line type, etc.

6. PLOS authors have the option to publish the peer review history of their article (what does this mean?). If published, this will include your full peer review and any attached files.

Reviewer #1: No

Reviewer #2: **Yes: **Yanhu Zhang

---

## [Author Response · Author response to Decision Letter 0]

17 Mar 2024

Dear Editor and reviewers,

Thank you for your letter and the reviewers’ comments on our manuscript entitled "Dynamic characteristics and fatigue life analysis of gyration platform of roadheader" (ID: PONE-D-23-35060). Those comments are very helpful for revising and improving our paper, as well as the important guiding significance to other research. We have studied the comments carefully and made corrections which we hope meet with approval. The main corrections are in the manuscript and the responds to the reviewers’ comments are as follows .

Replies to the reviewers’ comments:

Reviewer#1

Manuscript entitled "Dynamic characteristics and fatigue life analysis of gyration platform of roadheader" analyzed the cutting parameters of a roadheader during the excavation process. Some dynamic data was captured and discussed regarding the position and orientation or the roadheader. The authors believed that this study could provide a reference for the improvement of the gyration platform structure and the selection of a suitable swing speed. 

Special thanks to you for pointing out some important modifications needed in the manuscript. We have revised the manuscript according to your kind advises .

Concern # 1: The methods and the tools that the authors selected to carry out the research were too rough and simple. The authors are not familiar with the real working conditions of an excavator, there are many types for different working environments, to be straight forward and simple, coal cutter is coal cutter, roadheader is roadheader, one is for long wall mining face, one is for roadway excavation. The authors need to add more details of the differences, regarding the working conditions of them, rather than mix them together, sometimes declare that the roadheader is also adaptable for long wall mining face, which is clearly not precise.

Author response: 

Thank you very much for your comments. The roadheader is mainly used in tunnel excavation and coal mine tunnel excavation work. Based on the comments of the reviewers, we have revised the content of the introduction. 

The revisions are as follows:

The expeditious and secure excavation of coal mine roadway stands as a fundamental prerequisite for guaranteeing elevated coal production [1,2]. In the demanding context of coal mine roadway excavation, the cutting heads confront a milieu of complexity and harshness. This is exacerbated by the unstable physical and mechanical attributes of the cutting target (a heterogeneous coal-rock mass). 

Concern # 2: As stated by the authors, ‘In order to ensure the reliability of the gyration platform, when cutting in different swing directions, the swing speed should be reasonably selected based on the hardness of the coal and rock’. How did the authors consider the up and down movement, rather than just left and right, the real scenario for a roadheader is all directions, rather than just horizontal movement. Though it is acceptable to just discuss the horizontal one, it is indeed necessary to also consider the horizontal movement one, at least to add some discussions regarding the limitations of this study.

Author response: 

Thank you very much for your comments. In the conclusion section of the manuscript, we have added some discussions and explanations on the limitations of this study in the conclusion section.

The revisions are as follows:

In this study, we concentrated on scrutinizing the horizontal rotary motion of the roadheader. Nevertheless, during the practical cutting process, the roadheader exhibits both horizontal and vertical movements. The dynamic attributes of the roadheader exhibit variations in different directions of motion. Consequently, our study possesses certain limitations within its scope. To conduct a more comprehensive analysis of the roadheader's performance throughout the entire cutting process, our subsequent research will prioritize examining the dynamic characteristics of the roadheader during vertical motion.

Concern # 3: In section 2.1.3, the authors stated that the cutting head load was calculated by Matlab software, but the inner process is lacking, the purpose and the results from the calculation are also unknown, which both make this section rough and insufficient in details.

Author response: 

Thank you very much for your comments. Considering the structure and installation parameters of each pick, the force on an individual pick under specific conditions can be computed using Formulas 1-3. Subsequently, employing the calculation methodology outlined in Formulas 4-9, the external load experienced by the cutting head can be determined by converting the force on an individual pick to the center of mass of the cutting head. According to the reviewer's comments, we have added a flowchart for calculating the cutting head load. Based on this flowchart, we used MATLAB to achieve the calculation of the load.

The revisions are as follows:

In order to analyze the dynamic reliability of the roadheader, when analyzing the load of the cutting head under complex conditions, the rock samples to be excavated can be tested to obtain their physical and mechanical properties. Combined with the force analysis of the cutting pick, the cutting head load can be calculated. The calculation process is shown in Figure 4 (The Figure can be seen in the document responding to the reviewer's comments)..

Concern # 4: In section 2.1.3, the authors mentioned firmness coefficient, what was the strength values of the rock?

Author response: 

We updated the manuscript by adding the property parameters of rock samples according to the suggestions provided by Reviewer #1. Table 1 can be seen in the document responding to the reviewer's comments.

.

The revisions are as follows:

Based on the above analysis, a program was developed to calculate the cutting head load of the roadheader. The operational parameters were determined based on the geological conditions specific to this model's mining face. The properties of rock samples cut by the given roadheader were determined, and the results of the property measurements were presented in Table 1. When calculating the load, the average of two measurement results was taken for the computation. The cutting motor power is 75 kW, and the cutting head rotation speed is set at 53 r/min. Detailed parameters of the cutting head are provided in Table 2. Operating at full load, the maximum transverse swing speed of the cutting head is approximately 1.1 m/min. By executing the program, the instantaneous load of the cutting head can be computed for two working conditions: transverse swing from left to right and transverse swing from right to left. The resulting cutting head load curves are illustrated in Figure 6.

Concern # 5: Also in section 2.1.3, how did the authors obtain the data? it is a necessity to briefly introduce the methods, or the arrangement of the sensors that captured all data.

Author response: 

Thank you very much for your comments, and we have added a description of the data collection in the experiment and revised Figure 10 (The Figure can be seen in the document responding to the reviewer's comments).

The revisions are as follows:

The torque during the cutting process was obtained by installing a torque sensor on the cutting motor torque shaft, and the torque value was obtained using a data acquisition device.

Concern # 6: The most serious flaw of this paper exists in Fig. 8, obviously the cutting test couldn't shape a rectangular groove as shown in the figure, the cutting head was an overall domed appearance, how did it possible for a domed head cut a rectangular groove?

Author response: 

Thank you very much for your comments. This experiment was conducted at the Mining Hydraulic Technology and Equipment Engineering Research Center, equipped for testing and analyzing the dynamic performance of mining machinery, including shearers and roadheaders, as shown in Figure (a) . The simulated coal wall employed in this study serves for cutting experiments with roadheaders, as well as for experiments involving the cutting of the top and bottom rock layers by coal shearers, as shown in Figure (b) . The rectangular groove depicted in Figure 9 ( Revised as Figure 10 ) is a remnant from a prior shearer cutting test. Figure (a) and Figure (b) can be seen in the document responding to the reviewer's comments.

Reviewer#2

Concern # 1: The fatigue lifespan expression and/or result are not enough in the current manuscript that is not consist with the defined TITLE.

Author response: 

Thank you very much for your comments. This article investigates the dynamic characteristics of the gyration platform during lateral cutting, including the stress, deformation, and fatigue life of the gyration platform under different swing directions and speeds. Different lateral directions and swinging speeds have important effects on the stress distribution and deformation of the gyration platform. Under the premise of determining the cutting object, this article establishes the correlation between the fatigue life of the gyration platform and the horizontal swing speed. If an in-depth analysis is conducted on the fatigue life of the gyration platform, it is necessary to study the effects of different rock properties, cutting head speed, horizontal swing speed, and up and down swing speed on the fatigue life of the gyration platform. Thank you very much for your comments. In future research, we will conduct a more in-depth study on the fatigue life of the gyration platform.

Concern # 2: The analysis reference is not referring the classical strength theory despite the manuscript filling with enough stress and strain maps.

Author response: 

 Thank you very much for your comments. We have added the classical strength theory and corresponding references. Formula 19 can be seen in the document responding to the reviewer's comments

The revisions are as follows:

According to the Miner's Rule, the load acting on the gyration platform is composed of m different stresses[22]. The number of cycles at each stress level is denoted as n1, n2, ..., nm, and the corresponding fatigue life is represented as N1, N2, ..., Nm. Following the linear accumulation principle, the total damage to the rotary platform is calculated according to Formula 19.

When Ds=1, the distortion energy density of the gyration platform reaches its limit and fails.

Concern # 3: The current English expression need to improve and polish.

Author response: 

Modifications have been made to address the inappropriate grammar statements in the paper.

Concern # 4: The figures are low resolution without scope of academic publish standards.

Author response: 

 Thank you very much for your comments. We have revised the images and increased their resolution.

Concern # 5: The fitting curves are low in physical recombination without enough factors illustration in engineering applications. Why it is high-order function with the abscissa? (Fig. 7 and Fig. 17)

During the data fitting process, we employed polynomial fitting and Gaussian fitting methods and selected the most suitable fitting formula. Taking Figure 17 ( Revised as Figure 18) as an example, it depicts the deformation of the gyration platform with varying lateral swing speeds. By utilizing Gaussian fitting, the following results were obtained:

Formula (1) can be seen in the document responding to the reviewer's comments (1)

The R2 of the formula (1) is 0.9798.

Formula (2) can be seen in the document responding to the reviewer's comments (2)

The R2 of the formula (2) is 0.9995.

Similarly, polynomial fitting was applied to the data, revealing that this approach not only ensured an R2 value approaching unity but also provided a more straightforward and intuitive depiction of the relationship between the rotary platform's deformation and the lateral swing speed. We ultimately adopted polynomial fitting, and the formula (3) obtained from the fitting is:

Formula (3) can be seen in the document responding to the reviewer's comments (3)

The R2 of the formula (3) is 0.9999.

Concern # 6: Figure 1 and Figure 2 should add the 3D model to support the agreed assumption before the raised question.

Author response: 

According to the reviewer's comments, we have added a 3D model.

Concern # 7: The line diagrams in the referred figures should be normalized for a uniform style, including the font, size, line type, etc.

Author response: 

According to the reviewer's comments, we have revised the figures to ensure that they have a a uniform style, including the font, size, line type, etc.

Once again, thank you very much for your constructive comments and suggestions which would help us both in English and in depth to improve the quality of the paper.

Best regards

Sincerely,

Zhen Tian

E-Mail:lntutian2008@126.com

---

## [Decision Letter · Decision Letter 1]

10 May 2024

PONE-D-23-35060R1Dynamic characteristics and fatigue life analysis of gyration platform of roadheaderPLOS ONE

Dear Dr. Tian,

Thank you for submitting your manuscript to PLOS ONE. After careful consideration, we feel that it has merit but does not fully meet PLOS ONE’s publication criteria as it currently stands. Therefore, we invite you to submit a revised version of the manuscript that addresses the points raised during the review process.

Please, address all the comments made by the reviewers. 

We look forward to receiving your revised manuscript.

Kind regards,

Antonio Riveiro Rodríguez, PhD

Academic Editor

PLOS ONE

Reviewers' comments:

Reviewer's Responses to Questions

**Comments to the Author**

1. If the authors have adequately addressed your comments raised in a previous round of review and you feel that this manuscript is now acceptable for publication, you may indicate that here to bypass the “Comments to the Author” section, enter your conflict of interest statement in the “Confidential to Editor” section, and submit your "Accept" recommendation.

Reviewer #2: All comments have been addressed

Reviewer #3: All comments have been addressed

Reviewer #4: (No Response)

2. Is the manuscript technically sound, and do the data support the conclusions?

Reviewer #2: Yes

Reviewer #3: Yes

Reviewer #4: Partly

3. Has the statistical analysis been performed appropriately and rigorously? 

Reviewer #2: No

Reviewer #3: Yes

Reviewer #4: No

4. Have the authors made all data underlying the findings in their manuscript fully available?

Reviewer #2: No

Reviewer #3: Yes

Reviewer #4: No

5. Is the manuscript presented in an intelligible fashion and written in standard English?

Reviewer #2: Yes

Reviewer #3: Yes

Reviewer #4: Yes

6. Review Comments to the Author

Reviewer #2: Thanks for your revision. Some minor concerns are provided as follows:

(1) Deformation trend of the gyration platform in Fig 16 is unscientific because of the missed quantitative information, neither the legend, lined maps nor comparison.

(2) Some figures (e.g. Fig 17~Fig 20, Fig.22) are lacking in depth for optimized significance and designed objects.

(3) Results and Discussion section is raw and basic, especially there is no discussion and comparison analysis. Some constructive discussion is necessary for one research paper, rather than a one-plank narrative story.

(4) Three-lines table is welcome in the research paper.

Reviewer #3: The dynamic characteristics and fatigue life of the gyration platform of road header when cutting coal and rock, are developed in this manuscript. The simulated load was applied to the multibody model of the cutting section and dynamic simulation was conducted. It was found that the maximum stress and deformation of the gyration platform have a significant increase trend with the increase of swing speed. This paper is well-structured and of interest to this journal. But some minor issues need to be clarified before publication can be recommended.

1、“Due to the cutting power of this type of roadheader being 75kW, the swing speed should be ensured to be within 1.1m/min during the cutting process.” Please confirm if 1.1 m/min is correct？

2、“The maximum relative error of the effective torque values obtained by the two methods under different conditions is 10.2929%.” Please explain whether it is reasonable for the error to reach 10.2929%, and suggest modifying the model to reduce the error.

3、Figure 11 shows periodic phenomenon in acceleration Y, please explain?

Reviewer #4: Some calculations and numerical simulations were conducted to obtain the stress and fatigue life of a roadheader's cutting head and gyration platform. However, the siginification of this study fails to be stated clearly. Almost all the calculations were derived from some formulas cited from other literatures. Little innovative method was proposed. Some of my concerns are listed below.

1. The difficult or urgency for investigating the stress and fatigue life of a roadheader's gyration platform need to be explain firstly. Why does few of the existing studies involve the dynamics of the gyration platform?

2. The current studies cannot well illustrate the dynamic characteristics and fatigue life of gyration platform. Only the torque applied on the gyration platform and the acceleration and stress at some nodes were discussed. These data cannot reflect the dynamic characteristics such as vibration and modality. The fatigue life was figured out from a S-N curve of common Q235A material in the software by importing the stress distribution. Whether is this computation reliable to determine the actual fatigue life of the whole gyration platform?

3. In the sentence 'the cutting picks experience forces as illustrated in Figure 1', Figure 1 should be Figure 2.

4. Fig.16 should be presented at a more appropriate view angle to better illustrate the deformation. In addition, what does the color in this figure indicate? A color scale should be added next to the model.

5. What does the deformation in Fig.18 stand for? Where is this deformation?

6. What does the fatigue life in Fig.23 stand for? I just wonder what does the minimum fatigue life mean? If the nodes within the gyration platform had diffferent fatigue life, what should be the fatigue life of this component, i.e. gyration platform?

7. PLOS authors have the option to publish the peer review history of their article (what does this mean?). If published, this will include your full peer review and any attached files.

Reviewer #2: **Yes: **ZHANG Yanhu

Reviewer #3: No

Reviewer #4: No

---

## [Author Response · Author response to Decision Letter 1]

13 Jun 2024

Original Manuscript ID: PONE-D-23-35060R1

Original Article Title: Dynamic characteristics and fatigue life analysis of gyration platform of roadheader 

To: PLOS ONE

Re: Response to reviewers

Dear Editor and reviewers,

Thank you for your letter and the reviewers’ comments on our manuscript entitled "Dynamic characteristics and fatigue life analysis of gyration platform of roadheader" (ID: PONE-D-23-35060R1). Those comments are very helpful for revising and improving our paper, as well as the important guiding significance to other research. We have studied the comments carefully and made corrections which we hope meet with approval. The main corrections are in the manuscript and the responds to the reviewers’ comments are as follows .

Replies to the reviewers’ comments:

Reviewer#2

Special thanks to you for pointing out some important modifications needed in the manuscript. We have revised the manuscript according to your kind advises .

Concern # 1: Deformation trend of the gyration platform in Fig 16 is unscientific because of the missed quantitative information, neither the legend, lined maps nor comparison.

Author response: 

Thank you very much for your comments. We have revised Fig 16 by adding legend and color scale to better illustrate the deformation trend of gyration platform.

Concern # 2: Some figures (e.g. Fig 17~Fig 20, Fig.22) are lacking in depth for optimized significance and designed objects

Author response: 

Thank you very much for your comments. From Fig.17-Fig.22, we can observe the deformation and stress distribution of the gyration platform, and demonstrate the relationship between deformation and stress with the swing speed. As shown in the figures, we can see that the maximum deformation of the gyration platform mainly occurs at the ear connecting the gyration platform and the lifting hydraulic cylinder, and the maximum stress area is mainly distributed around the bolt holes of the rotary bearings. The above analysis results provide a certain reference for the structural optimization of the gyration platform and the selection of rotation speed. In order to further improve the reliability of the gyration platform, we have added structural improvement measures. Through Workbench, we have optimized parameters P10 (thickness of the two outer rib plates ), P2 (thickness of the two inner rib plates), P5 (thickness of the upper ear connecting rib plates), and P11 (radius of the bolt hole connected to the rotary bearing), as shown in Figure 24.

The revisions are as follows:

The locations with high stress on the gyration platform are distributed at the bolt holes connected to the rotary bearings at the front of the gyration platform, the middle part of the connecting rib plate of the ear connected to the rotary hydraulic cylinder, the ear connected to the lifting hydraulic cylinder, and the ear connected to the cutting motor box. As shown in Figure 22, these locations are also areas with lower fatigue life. In order to reduce the maximum stress of the gyration platform during the cutting process and improve its fatigue life, the parameters P10 (thickness of the two outer rib plates FD1), P2 (thickness of the two inner rib plates), P5 (thickness of the upper ear connecting rib plates), and P11 (half diameter of the bolt hole connected to the rotary bearing) were optimized, as shown in Figure 24.

According to the design dimensions of the cutting machine case and the relative position between the body frame and the hydraulic cylinder, the variation range of parameters P10, P2, and P5 is 25-30mm, and the variation range of parameter P11 is 8.1-9.9mm. The optimization design objective parameters are to minimize the increase in mass of the gyration platform and reduce the maximum stress of the gyration platform. Figure 25 shows the optimized design nodes of different parameters obtained from simulation analysis.

20 sampling points for optimized design were determined based on the numerical range of design parameters, and the MOGA method was selected for optimization based on model characteristics. The simulation provides three optimization schemes, as shown in Table 6, and by comparing the relevant parameters in different schemes, scheme 1 was selected as the optimization scheme. The mass of the optimized model is 749.18 kg, representing a mere 9.03 kg increase compared to its pre-optimized state. Furthermore, the maximum stress in the optimized model measures 293.23 MPa, indicating a substantial reduction of 71.34 MPa from the pre-optimization stress level of 364.57 MPa. These analyses underscore a noteworthy enhancement in the stress condition of the optimized gyration platform.

Concern # 3: Results and Discussion section is raw and basic, especially there is no discussion and comparison analysis. Some constructive discussion is necessary for one research paper, rather than a one-plank narrative story.

Author response: 

Thank you very much for your comments, We have revised the conclusion.

The revisions are as follows:

Based on the analysis of individual cutting pick forces, a transformation of all cutting forces to the centroid of the cutting head was executed via coordinate transformation. This methodology effectively centralized the forces, facilitating a comprehensive analysis of the instantaneous loads acting on the cutting head under diverse lateral swing conditions. The resultant loads served as external excitations for the dynamic simulation of the cutting unit model. 

Noteworthy insights were gleaned through dynamic simulation analysis of the rigid-flexible coupled multibody model of the roadheader's cutting part. It was discerned that the vibration acceleration of the cutting head in the Z direction is greater than the other two directions. The connecting ear of nodes 62796 and 62797 on the gyration platform vibrates the most violently, and the vibration acceleration in the Y direction is greater than the other two directions. At the same time, this position is also the location where the gyration platform undergoes significant deformation. The gyration platform experiences a maximum stress of 258.71MPa during right-to-left lateral swinging cutting, while during left-to-right lateral swinging cutting, the maximum stress escalates to 364.57MPa. It was discerned that, during left-to-right swing cutting, the gyration platform experienced markedly elevated maximum stress in comparison to right-to-left swing cutting. This heightened stress was predominantly localized in the vicinity of the bolt holes interconnecting the gyration platform and the rotary bearings. The fatigue life analysis shows that the node with the shortest fatigue life is mainly located in the front area of the rotating platform, near the edge of the bolt hole connecting the rotating platform and the rotating bearing. Among these nodes, node 33 exhibits the lowest fatigue life and can withstand a maximum cyclic load of 5.366×106 cycles. With an escalation in swing speed, there was a conspicuous intensification in both the maximum stress and deformation of the gyration platform, consequently leading to a marked reduction in its fatigue life. It is noteworthy that, when confronted with the challenge of cutting hard rock,it is necessary to reduce the rotational speed to ensure the reliability of key components. In order to improve the reliability of the gyration platform, the relevant structural parameters of the gyration platform, such the thickness of the two outer rib plates, the thickness of the two inner rib plates, the thickness of the upper ear connecting rib plates, and the diameter of the bolt hole connected to the rotary bearing, were optimized and designed. The maximum stress of the optimized model is 293.23MPa, which is 71.34 MPa lower than the 364.57 MPa before optimization. The analysis results indicate that the stress state of the optimized gyration platform can be significantly improved.

The research findings not only furnish valuable guidance for enhancing the structural integrity of the gyration platform but also provide insights into the judicious selection of swing speeds, thereby contributing significantly to the overall enhancement of roadheader performance. In this study, we concentrated on scrutinizing the horizontal rotary motion of the roadheader. Nevertheless, during the practical cutting process, the roadheader exhibits both horizontal and vertical movements. The dynamic attributes of the roadheader exhibit variations in different directions of motion. Consequently, our study possesses certain limitations within its scope. To conduct a more comprehensive analysis of the roadheader's performance throughout the entire cutting process, our subsequent research will prioritize examining the dynamic characteristics of the roadheader during vertical motion.

Concern # 4: Three-lines table is welcome in the research paper.

Author response: 

Thank you very much for your comments, We have revised the tables.

Reviewer#3

Special thanks to you for pointing out some important modifications needed in the manuscript. We have revised the manuscript according to your kind advises .

Concern # 1: “Due to the cutting power of this type of roadheader being 75kW, the swing speed should be ensured to be within 1.1m/min during the cutting process.” Please confirm if 1.1 m/min is correct？

Author response: 

Thank you very much for your comments. According to formulas 1 and 2 (Ref : Li G. X. & Li X. H. Design of Coal Mining Machinery,Shenyang: Liaoning University Press; 1994.), the swing speed of the roadheader under the condition of maximum cutting power can be calculated. It is correct to calculate a maximum swing speed of approximately 1.1 m/min when the cutting power is 75kW.

The cutting power could be obtained from the following formula (1):

In the formula above: Vp is the swing speed of the working mechanism; is the mechanical efficiency of the transmission device. Fn is the average total cutting resistance of spiral drum, which can be obtained from the following formula (2):

In the formula above: KYK is the coefficient of influence of the thickness of the mined coal seam on the envelope angle formed by the working mechanism and the average cutting force; Kcc is the cutting direction coefficient of coal seam weakening and working mechanism turning relative to coal bedding; NC is the number of tooth groups; Zcpi is the average cutting force of the i-th cut; ni is the number of cutting teeth working simultaneously.

Concern # 2: “The maximum relative error of the effective torque values obtained by the two methods under different conditions is 10.2929%.” Please explain whether it is reasonable for the error to reach 10.2929%, and suggest modifying the model to reduce the error.

Author response: 

Thank you very much for your comments. By checking the model, it was found that when the swing speed from left to right is 0.9 m/min, the simulated torque value should be 1.1063, resulting in an error of 9.48%. In the cutting test, we used cement, water reducing agent, and lime to create a simulated working face for the roadheader to cut. The materials produced in this way may have certain non-uniform properties. In simulation calculations, the theoretical calculation formula is calculated based on homogeneous rock materials, which may cause errors between the two methods. However, through comparative analysis, it was found that the minimum error of the two methods is 3.66%, the maximum error is 9.48%, and the error is within 10%. Therefore, it can be considered that the simulation results can reflect the actual working situation of the roadheader.

The revisions are as follows:

To validate the accuracy of the load calculations, a simulated working face mirroring the properties of the rock cut by the specified roadheader was created using cement, water reducing agent, and lime. The cutting part structure of the roadheader was replicated for cutting tests on this simulated working face, as illustrated in Figure 10. The torque during the cutting process was obtained by installing a torque sensor on the cutting motor torque shaft, and the torque value was obtained using a data acquisition device.The results of tests and simulations comparing torque with varying swing speeds are presented in Figure 11 and Table 3. The maximum relative error between the effective torque values obtained through experimental and simulation methods under different conditions is 9.48%. This indicates the reliability of the load data derived from the theoretical calculations. Importantly, the calculated load data serves as a robust source of external excitation for multi-body dynamics simulations.

Concern # 3: Figure 11 shows periodic phenomenon in acceleration Y, please explain?

Author response: 

Thank you very much for your comments. The vibration acceleration mentioned by the reviewer should be a periodic phenomenon observed in the vibration acceleration shown in Figures 13 and 15. When the cutting head cuts the rock, the number of the cutting picks involved in cutting is constantly changing, and it shows a periodic change. As a result, the load on the cutting head also changes periodically, as shown in Figure 6. when the load was added to the model and simulated, the acceleration in the three directions of the cutting head and the node 62796 of gyration platform will exhibit periodic phenomenon.

Reviewer#4

Special thanks to you for pointing out some important modifications needed in the manuscript. We have revised the manuscript according to your kind advises .

Concern # 1: The difficult or urgency for investigating the stress and fatigue life of a roadheader's gyration platform need to be explain firstly. Why does few of the existing studies involve the dynamics of the gyration platform?

Author response: 

 Thank you very much for your comments. We have added a necessity analysis for studying the stress and fatigue life of the gyration platform in the introduction section.The research on the dynamic characteristics of roadheader mainly focuses on the analysis of the forces acting on the cutting teeth and the impact of cutting teeth on the vibration of the roadheader. Many existing studies have also been conducted on the gyration platform, but static or modal analysis methods are often used, which cannot fully obtain the dynamic characteristics of the gyration platform during the cutting process of the roadheader. We found that studying the stress, deformation, and fatigue life of the gyration platform under different lateral directions and lateral velocities can help us find methods to improve the reliability of the gyration platform and thereby improve the performance of the roadheader.

The revisions are as follows:

These studies have confirmed the accuracy and reliability of finite element and dynamic simulation results, yielding valuable outcomes. For the roadheader, the gyration platform as a vital component, linking the main body frame and supporting the cutting arm. During the cutting process, the gyration platform needs to achieve the lifting and rotation of the cutting arm and bear strong alternating loads, which is a weak link. However, few of the existing studies have involved the dynamics of the gyration platform under various lateral swing conditions and the influence of the lateral swing speed on the fatigue performance.

Concern # 2: The current studies cannot well illustrate the dynamic characteristics and fatigue life of gyration platform. Only the torque applied on the gyration platform and the acceleration and stress at some nodes were discussed. These data cannot reflect the dynamic characteristics such as vibration and modality. The fatigue life was figured out from a S-N curve of common Q235A material in the software by importing the stress distribution. Whether is this computation reliable to determine the actual fatigue life of the whole gyration platform?

Author response: 

Thank you very much for your comments. In order to better illustrate the dynamic characteristics of the gyration platform, we conducted modal analysis on the cutting part of the roadheader, obtained the first eight modal frequencies andvibration modal shape of the gyration platform, and analyzed the vibration deformation of the gyration platf

---

## [Decision Letter · Decision Letter 2]

7 Aug 2024

PONE-D-23-35060R2Dynamic characteristics and fatigue life analysis of gyration platform of roadheaderPLOS ONE

Dear Dr. Tian,

Thank you for submitting your manuscript to PLOS ONE. After careful consideration, we feel that it has merit but does not fully meet PLOS ONE’s publication criteria as it currently stands. Therefore, we invite you to submit a revised version of the manuscript that addresses the points raised during the review process.

 Please, try to address the comments and concerns of the reviewers. 

We look forward to receiving your revised manuscript.

Kind regards,

Antonio Riveiro Rodríguez, PhD

Academic Editor

PLOS ONE

Journal Requirements:

Reviewers' comments:

Reviewer's Responses to Questions

**Comments to the Author**

1. If the authors have adequately addressed your comments raised in a previous round of review and you feel that this manuscript is now acceptable for publication, you may indicate that here to bypass the “Comments to the Author” section, enter your conflict of interest statement in the “Confidential to Editor” section, and submit your "Accept" recommendation.

Reviewer #4: All comments have been addressed

2. Is the manuscript technically sound, and do the data support the conclusions?

Reviewer #4: Partly

3. Has the statistical analysis been performed appropriately and rigorously? 

Reviewer #4: N/A

4. Have the authors made all data underlying the findings in their manuscript fully available?

Reviewer #4: Yes

5. Is the manuscript presented in an intelligible fashion and written in standard English?

Reviewer #4: Yes

6. Review Comments to the Author

Reviewer #4: The manuscript has been revised carefully. But I still wonder the academical significance for investigating fatigue life of a roadheader's cutting head and gyration platform. Neither innovative methods nor valuable conclusions were proposed. In addition, I don't think the fatigue life has been well examined in this study, because the 'fatigue life' was just given from a certain stress according to S-N curves of materials. It means the original data were still stress, i.e. the discusses were still stress analysis. Only difference is that they were explained from another view by assigning them to a certain fatigue life.

Nevertheless, this manuscript presents with relative high scholarly format and thus I'm not opposed to accepting it after minor modification. The conclusion need to be revised. It's too verbose to well refelct the objective and some main contribution of this study.

7. PLOS authors have the option to publish the peer review history of their article (what does this mean?). If published, this will include your full peer review and any attached files.

Reviewer #4: No

---

## [Author Response · Author response to Decision Letter 2]

9 Aug 2024

Dear Editor and reviewers,

Thank you for your letter and the reviewers’ comments on our manuscript entitled "Dynamic characteristics and fatigue life analysis of gyration platform of roadheader" (ID: PONE-D-23-35060R2). Those comments are very helpful for revising and improving our paper, as well as the important guiding significance to other research. We have studied the comments carefully and made corrections which we hope meet with approval. The main corrections are in the manuscript and the responds to the reviewers’ comments are as follows .

Replies to the reviewers’ comments:

Reviewer#4

Special thanks to you for pointing out some important modifications needed in the manuscript. We have revised the manuscript according to your kind advises .

Concern # 1: The manuscript has been revised carefully. But I still wonder the academical significance for investigating fatigue life of a roadheader's cutting head and gyration platform. Neither innovative methods nor valuable conclusions were proposed. In addition, I don't think the fatigue life has been well examined in this study, because the 'fatigue life' was just given from a certain stress according to S-N curves of materials. It means the original data were still stress, i.e. the discusses were still stress analysis. Only difference is that they were explained from another view by assigning them to a certain fatigue life.

Nevertheless, this manuscript presents with relative high scholarly format and thus I'm not opposed to accepting it after minor modification. The conclusion need to be revised. It's too verbose to well refelct the objective and some main contribution of this study.

Author response: 

Thank you very much for your comments. 

The roadheader will be subjected to nonlinear impact loads during the process of cutting rocks, which will have adverse effects on the reliability of the key mechanical structure of the roadheader. Nonlinear impact load is the result of the cutting teeth on the cutting head breaking the rock, so studying the impact load on the cutting head is a prerequisite for studying the reliability and fatigue life of the mechanical structure of the roadheader. For a roadheader, the gyration platform is an important component that connects the main frame and supports the cutting arm. During the cutting process, the gyration platform needs to achieve the lifting and rotation of the cutting arm and withstand strong alternating loads, which is a weak link. If the gyration platform is damaged, it will affect the normal progress of excavation work. Therefore, the reliability and fatigue life of the gyration platform are crucial for the normal operation of the roadheader. On the basis of studying the impact load on the cutting head under different working conditions, this paper analyzes the dynamic characteristics and fatigue life of the gyration platform, which is of great significance for the structural design of the gyration platform and the reasonable selection of the swinging speed of the roadheader.

The gyration platform operates in a highly complex vibration environment, where various parts of its structure are subjected to stress and strain that change over time. Studying the fatigue life of the gyration platform is to investigate the number of stress cycles experienced by the gyration platform before fatigue failure. The commonly used fatigue life estimation method in engineering currently involves first determining the external load and obtaining the S-N curve of the material or component through testing to describe the relationship between the number of fatigue fracture cycles and cyclic stress that the material is subjected to. At the same time, the Miner's fatigue damage theory was used to estimate the fatigue life of the gyration platform. The Miner's fatigue damage theory has the characteristics of fast calculation speed, easy to understand solution process, and in most cases, its estimated life value is quite close to the experimental results. Therefore, the Miner's fatigue damage theory was used in this study. In Miner's fatigue damage theory, the stress of the gyration platform is the basis for analyzing fatigue life. Only by determining the stress of the gyration platform and establishing an accurate load spectrum can the accurate analysis of the fatigue life of the gyration platform be achieved. 

According to the reviewer's comments, we have revised the conclusion.

The revisions are as follows:

Based on the analysis of individual cutting pick forces, a transformation of all cutting forces to the centroid of the cutting head was executed via coordinate transformation. This methodology effectively centralized the forces, facilitating a comprehensive analysis of the instantaneous loads acting on the cutting head under diverse lateral swing conditions. The resultant loads served as external excitations for the dynamic simulation of the cutting unit model. 

Noteworthy insights were gleaned through dynamic simulation analysis of the rigid-flexible coupled multibody model of the roadheader's cutting part. During left-to-right swing cutting, the gyration platform experienced markedly elevated maximum stress in comparison to right-to-left swing cutting. This heightened stress was predominantly localized in the vicinity of the bolt holes interconnecting the gyration platform and the rotary bearings. The fatigue life analysis shows that the node with the shortest fatigue life is mainly located in the front area of the rotating platform, near the edge of the bolt hole connecting the rotating platform and the rotating bearing. With an escalation in swing speed, there was a conspicuous intensification in both the maximum stress and deformation of the gyration platform, consequently leading to a marked reduction in its fatigue life. It is noteworthy that, when confronted with the challenge of cutting hard rock,it is necessary to reduce the swing speed to ensure the reliability of key components. 

In order to improve the reliability of the gyration platform, the relevant structural parameters of the gyration platform, such the thickness of the two outer rib plates, the thickness of the two inner rib plates, the thickness of the upper ear connecting rib plates, and the diameter of the bolt hole connected to the rotary bearing, were optimized and designed. The analysis results indicate that the stress state of the optimized gyration platform can be significantly improved. The research findings not only furnish valuable guidance for enhancing the structural integrity of the gyration platform but also provide insights into the judicious selection of swing speeds, thereby contributing significantly to the overall enhancement of roadheader performance.

Once again, thank you very much for your constructive comments and suggestions which would help us improve the quality of the paper.

Best regards

Sincerely,

Zhen Tian

E-Mail:lntutian2008@126.com

---

## [Decision Letter · Decision Letter 3]

27 Sep 2024

PONE-D-23-35060R3Dynamic characteristics and fatigue life analysis of gyration platform of roadheaderPLOS ONE

Dear Dr. Tian,

Thank you for submitting your manuscript to PLOS ONE. After careful consideration, we feel that it has merit but does not fully meet PLOS ONE’s publication criteria as it currently stands. Therefore, we invite you to submit a revised version of the manuscript that addresses the points raised during the review process. Please, address all the comments made by the reviewers. 

We look forward to receiving your revised manuscript.

Kind regards,

Antonio Riveiro Rodríguez, PhD

Academic Editor

PLOS ONE

Journal Requirements:

Reviewers' comments:

Reviewer's Responses to Questions

**Comments to the Author**

1. If the authors have adequately addressed your comments raised in a previous round of review and you feel that this manuscript is now acceptable for publication, you may indicate that here to bypass the “Comments to the Author” section, enter your conflict of interest statement in the “Confidential to Editor” section, and submit your "Accept" recommendation.

Reviewer #3: All comments have been addressed

Reviewer #5: All comments have been addressed

Reviewer #6: (No Response)

2. Is the manuscript technically sound, and do the data support the conclusions?

Reviewer #3: Yes

Reviewer #5: Yes

Reviewer #6: Yes

3. Has the statistical analysis been performed appropriately and rigorously? 

Reviewer #3: (No Response)

Reviewer #5: Yes

Reviewer #6: N/A

4. Have the authors made all data underlying the findings in their manuscript fully available?

Reviewer #3: Yes

Reviewer #5: (No Response)

Reviewer #6: No

5. Is the manuscript presented in an intelligible fashion and written in standard English?

Reviewer #3: Yes

Reviewer #5: Yes

Reviewer #6: Yes

6. Review Comments to the Author

Reviewer #3: This manuscript provides a comprehensive and insightful analysis of the dynamic characteristics and fatigue life of the gyration platform in roadheaders under nonlinear impact loads. The authors have conducted a rigorous study by integrating modal analysis, stress-strain evaluations, and Miner's fatigue damage theory, leading to valuable conclusions regarding structural optimization and swing speed selection. The methodology is sound, and the results are significant for enhancing the reliability and operational efficiency of roadheaders. The paper is well-written and clearly presented. I strongly recommend the acceptance of this manuscript for publication.

Reviewer #5: Dear Editor,

Thank you for the opportunity to review the research paper submitted for potential publication in PLOS ONE. The manuscript has been evaluated for both its scientific merit and linguistic precision.

This is the third round of revisions, based on the referees' comments from previous rounds, in which I was not involved. However, it appears that the authors have adequately addressed all the feedback provided. The paper presents a study investigating the dynamic characteristics and fatigue life of the roadheader's gyration platform during the cutting process. The importance of exploring these aspects is well-recognized, and the subject matter fits within the journal’s scope.

The research background, originality, contribution to the field, and methodology are appropriate. While the paper references around 23 scientific sources, both the introduction and discussion sections would benefit from a more extensive integration of relevant previous studies, and the number of citations should be increased accordingly.

Regarding graphics and tables: I recommend using more professional graphic design techniques to enhance the quality of graphs, schematics, and tables. Ensure consistency in font sizes, symbols, and units between figures, tables, and the main text to improve legibility. In particular, revise the FEA printouts and ensure that the legends for each analysis are clearly readable.

In conclusion, I recommend this paper for publication in PLOS ONE after minor revisions.

Yours sincerely,

Reviewer #6: Comments and observations:

1. Formula (1) – (12) should be provided with units. How is the term “width of the pick blade” (formula (1)) understood if the cutting head is equipped with conical picks? Where is the pick width measured and how much is it – the tip diameter (Alloy head diameter) is given in Tab. 2)?

2. What are the values given in formulas (17) and (18) (187.9 mm and 2.2 mm/s), on what basis were they calculated, since no data is given?

3. Table 2 – the pick shank diameter and alloy head diameter seem to be unnecessary from the point of view of the presented analyses.

4. The description of Fig. 7-9 states that with the increase in boom swing speed, the torque and power consumption for cutting increase. Meanwhile, Fig. 9 shows that the cutting torque decreases. Where in Fig. 7 and 8 is the increase in load and power visible – the courses have a periodic character with a rotation period of the cutting head (1.13 s)?

5. The last line of the paragraph under Fig. 6 refers to optimal mining conditions. This is a rather lofty formulation, and it is not clear what it means.

6. In the description of the directions of acceleration components in the paragraph under Fig. 13, the Z axis is the vertical direction, and the X axis is the horizontal direction. This is different from Figs. 2 and 3. This is confusing.

7. How were the curves shown in Figs. 13, 15, 17, and 20 obtained – is this the result of computer simulation with the excitation shown in Fig. 6?

8. Table 4 – there is “Rsm” – it should be “RMS”.

9. Fig. 21 – for clarity, the lines could be made in different colors.

7. PLOS authors have the option to publish the peer review history of their article (what does this mean?). If published, this will include your full peer review and any attached files.

Reviewer #3: No

Reviewer #5: No

Reviewer #6: No

---

## [Author Response · Author response to Decision Letter 3]

23 Oct 2024

Original Manuscript ID: PONE-D-23-35060R3

Original Article Title: Dynamic characteristics and fatigue life analysis of gyration platform of roadheader 

To: PLOS ONE

Re: Response to reviewers

Dear Editor and reviewers,

Thank you for your letter and the reviewers’ comments on our manuscript entitled "Dynamic characteristics and fatigue life analysis of gyration platform of roadheader" (ID: PONE-D-23-35060R3). Those comments are very helpful for revising and improving our paper, as well as the important guiding significance to other research. We have studied the comments carefully and made corrections which we hope meet with approval. The main corrections are in the manuscript and the responds to the reviewers’ comments are as follows .

Replies to the reviewers’ comments:

Reviewer#6

Special thanks to you for pointing out some important modifications needed in the manuscript. We have revised the manuscript according to your kind advises .

Concern # 1:

1. Formula (1) – (12) should be provided with units. How is the term “width of the pick blade” (formula (1)) understood if the cutting head is equipped with conical picks? Where is the pick width measured and how much is it – the tip diameter (Alloy head diameter) is given in Tab. 2)?

Author response: 

Thank you very much for your comments. We have added the units in the parameter descriptions below each formula. Based on the comments from the reviewers, we have revised the meaning of the relevant parameters in Formula 1. When the cutting head using flat picks, the parameter kb is required, and the accurate meaning of kb is the influence coefficient of the blade width of the flat pick. When cutting head using conical picks, the geometric shape influence coefficient kg=kφ·kφ’·kd. At this time, the influence coefficient of the cutting width of the cutting pick is not required to calculate the force of the cutting pick, but the influence coefficient of alloy head diameter is required to calculate the geometric shape influence coefficient kg. The influence coefficient of alloy head diameter kd can be interpolated based on the data in Table 1.

According to the coal breaking theory, the diameter of the alloy head d and the diameter of pick shank D can be measured according to the following Fig. a, where the calculated working width of the cutting pick is half of the diameter of pick shank.

Concern # 2:

2.What are the values given in formulas (17) and (18) (187.9 mm and 2.2 mm/s), on what basis were they calculated, since no data is given?

Author response: 

Thank you very much for your comments. By checking the model, we have added relevant data, which can be input into formula 17 to calculate the elongation of the hydraulic cylinder. The elongation of the hydraulic cylinder was calculated to be 187.9mm. According to the swing model shown in Figure 5, it can be calculated that when the maximum swing speed is 1.1m/min, the extension and contraction speed of the hydraulic cylinder is 2.2mm/s.

We have revised the corresponding content of the manuscript. The revisions are as follows:

(1) According to the structure and actual working conditions of this type roadheader, AA’=332.9mm, AB=646.7mm, OC=3000mm, L=1500mm, α=30°, β=58.3°, γ=54.6°. The elongation of the hydraulic cylinder was calculated to be 187.9mm. 

Based on the operational efficiency stipulated for the roadheader, the cutting time required for the yaw cutting can be computed. Subsequently, the extension speed of the hydraulic cylinder is determined by formula 18.

(2)Operating at full load, the maximum transverse swing speed of the cutting head is approximately 1.1 m/min. At this time, the expansion and contraction speed of the hydraulic cylinder is 2.2mm/s. 

Concern # 3:

3.Table 2 – the pick shank diameter and alloy head diameter seem to be unnecessary from the point of view of the presented analyses.

Author response:

Thank you very much for your comments. Based on the response to comment # 1 and the explanation of Formulas 1-3, when the cutting head equipped with conical picks, the calculation of the force on the cutting pick requires the diameter of the alloy head, and the diameter of the pick shank is not a necessary parameter. According the reviewer's comments, we have revised Table 2.

Concern # 4:

4.The description of Fig. 7-9 states that with the increase in boom swing speed, the torque and power consumption for cutting increase. Meanwhile, Fig. 9 shows that the cutting torque decreases. Where in Fig. 7 and 8 is the increase in load and power visible – the courses have a periodic character with a rotation period of the cutting head (1.13 s)?

Author response: 

Thank you very much for your comments. Fig. 7 and Fig. 8 shows the torque and power variation curves of the cutting head at a swinging speed of 1.1m/s. Due to the fact that the roadheader starts cutting rocks at 0.3s, the torque and power consumption of the cutting head increase sharply from 0.3s. Due to the rotation of the cutting head, the number of cutting teeth involved in cutting rocks on the cutting head is constantly changing. Therefore, the fluctuation of the curve in Fig.7 and Fig. 8 has a certain periodicity, with a fluctuation period of about 1.13s. Figure 9 shows the changes in torque and power under different swing speed conditions.

Concern # 5:

5.The last line of the paragraph under Fig. 6 refers to optimal mining conditions. This is a rather lofty formulation, and it is not clear what it means.

Author response: 

Thank you very much for your comments. The cutting motor power of this roadheader is 75 kW, and the rotation speed of the cutting head is 53 r/min. Based on the measurement results of rock properties, it is calculated and analyzed that the maximum transverse swing speed of the cutting head is approximately 1.1 m/min. During the cutting process, the swing speed must be kept within 1.1 m/min to ensure that the machine is not overloaded. According to the reviewer's comments, we have revised this sentence to make it more accurate.

Concern # 6:

6. In the description of the directions of acceleration components in the paragraph under Fig. 13, the Z axis is the vertical direction, and the X axis is the horizontal direction. This is different from Figs. 2 and 3. This is confusing.

Author response: 

Thank you very much for your comments. In the force analysis of cutting pick, it is customary to name cutting resistance Fz, traction resistance Fy, and lateral force Fx. Therefore, according to the different subscripts of the forces, the direction description of the coordinate system in force analysis is often consistent with the subscripts. In this manuscript, When adding the loads as excitation in dynamic simulation, we have already adjusted the direction of the loads based on the model's coordinate system direction in the software. The Z direction in Fig.13 is consistent with the X direction in Fig. 2, as shown in Fig. b. Therefore, although the coordinate systems in Fig. 13, Fig. 2 and Fig. 3 are different, the data in the corresponding directions are correct. According to the reviewer's comments, in order to ensure consistency between the two coordinate systems, we have adjusted the coordinate system of the simulation model, as shown in Fig. c, and the adjusted coordinate system is consistent with the coordinate systems in Fig. 2 and Fig. 3, the data in Fig. 13, Fig. 15, and Table 4 have been revised.

Concern # 7:

7. How were the curves shown in Figs. 13, 15, 17, and 20 obtained – is this the result of computer simulation with the excitation shown in Fig. 6?

Author response: 

Thank you very much for your comments. Based on the force calculation of the cutting pick and the force transformation of cutting head, we can calculate the loads of the cutting head with different motion conditions through a simulation program, as shown in Fig. 6. In the dynamic software ADAMS, these loads were added as external excitations to the centroid of the cutting head in the dynamic simulation of the roadheader. Through dynamic simulation, the vibration characteristics of the cutting head and the flexible gyration platform, the stress-strain information of the flexible gyration platform can be obtained. The vibration acceleration in Fig.13 and Fig.15, the deformation in Fig.17, and the stress variation in Fig. 20 were obtained through dynamic simulation.

Concern # 8:

8. Table 4 – there is “Rsm” – it should be “RMS”.

Author response: 

Thank you very much for your comments. We have revised “Rsm ”to “RMS”.

Concern # 9:

9. Fig. 21 – for clarity, the lines could be made in different colors.

Author response: 

Thank you very much for your comments, We have revised the Fig. 21 and changed the colors of the lines in Figure.

Once again, thank you very much for your constructive comments and suggestions which would help us improve the quality of the paper. We hope that these revisions fulfill the requirements to make the manuscript acceptable for publication in PLOS ONE.

Best regards

Sincerely,

Zhen Tian

E-Mail:lntutian2008@126.com

---

## [Decision Letter · Decision Letter 4]

11 Nov 2024

Dynamic characteristics and fatigue life analysis of gyration platform of roadheader

PONE-D-23-35060R4

Dear Dr. Tian,

We’re pleased to inform you that your manuscript has been judged scientifically suitable for publication and will be formally accepted for publication once it meets all outstanding technical requirements.

Kind regards,

Antonio Riveiro Rodríguez, PhD

Academic Editor

PLOS ONE

Additional Editor Comments (optional):

Reviewers' comments:

Reviewer's Responses to Questions

**Comments to the Author**

1. If the authors have adequately addressed your comments raised in a previous round of review and you feel that this manuscript is now acceptable for publication, you may indicate that here to bypass the “Comments to the Author” section, enter your conflict of interest statement in the “Confidential to Editor” section, and submit your "Accept" recommendation.

Reviewer #3: All comments have been addressed

2. Is the manuscript technically sound, and do the data support the conclusions?

Reviewer #3: Yes

3. Has the statistical analysis been performed appropriately and rigorously? 

Reviewer #3: Yes

4. Have the authors made all data underlying the findings in their manuscript fully available?

Reviewer #3: Yes

5. Is the manuscript presented in an intelligible fashion and written in standard English?

Reviewer #3: Yes

6. Review Comments to the Author

Reviewer #3: (No Response)

7. PLOS authors have the option to publish the peer review history of their article (what does this mean?). If published, this will include your full peer review and any attached files.

Reviewer #3: No

---

## [Editor Report · Acceptance letter]

4 Dec 2024

PONE-D-23-35060R4 

PLOS ONE

Dear Dr. Tian, 

I'm pleased to inform you that your manuscript has been deemed suitable for publication in PLOS ONE. Congratulations! Your manuscript is now being handed over to our production team.

Kind regards, 

on behalf of

Dr. Antonio Riveiro Rodríguez 

Academic Editor

PLOS ONE